# Molecular principles of redox-coupled sodium pumping of the ancient Rnf machinery

Anuj Kumar[1,2,5], Jennifer Roth [2,5], Hyunho Kim[3,5], Patricia Saura [3,5], Stefan Bohn [4], Tristan Reif-Trauttmansdorff[1], Anja Schubert[2], Ville R. I. Kaila [3] ✉, Jan M. Schuller [1] ✉ & Volker Müller [2] ✉

The Rnf complex is the primary respiratory enzyme of several anaerobic prokaryotes that transfers electrons from ferredoxin to NAD$^+$ and pumps ions (Na$^+$ or H$^+$) across a membrane, powering ATP synthesis. Rnf is widespread in primordial organisms and the evolutionary predecessor of the Na$^+$-pumping NADH-quinone oxidoreductase (Nqr). By running in reverse, Rnf uses the electrochemical ion gradient to drive ferredoxin reduction with NADH, providing low potential electrons for nitrogenases and CO$_2$ reductases. Yet, the molecular principles that couple the long-range electron transfer to Na$^+$ translocation remain elusive. Here, we resolve key functional states along the electron transfer pathway in the Na$^+$-pumping Rnf complex from *Acetobacterium woodii* using redox-controlled cryo-electron microscopy that, in combination with biochemical functional assays and atomistic molecular simulations, provide key insight into the redox-driven Na$^+$ pumping mechanism. We show that the reduction of the unique membrane-embedded [2Fe2S] cluster electrostatically attracts Na$^+$, and in turn, triggers an inward/outward transition with alternating membrane access driving the Na$^+$ pump and the reduction of NAD$^+$. Our study unveils an ancient mechanism for redox-driven ion pumping, and provides key understanding of the fundamental principles governing energy conversion in biological systems.

Cellular metabolism is powered by membrane-bound protein complexes that catalyse the generation of an electrochemical ion gradient across a biological membrane that drives the synthesis of adenosine triphosphate (ATP) and active transport[1]. In many anaerobic bacteria and archaea that grow at the thermodynamic limit of life, where oxidation of low energy substrates result in small driving forces[2], the generation of ion gradients relies on the membrane-bound Rnf complex[3]. The *rnf* genes were first described in *Rhodobacter capsulatus*[4] in which they encode for a membrane-bound enzyme complex driving nitrogen (N$_2$) fixation, and similar genes have been found in many prokaryotic genomes[3]. *Acetobacterium woodii* is the model organism of a group of strictly anaerobic acetogenic bacteria, which grow by reduction of CO$_2$ to acetate with electrons derived from molecular hydrogen. *A. woodii* couples the CO$_2$ fixation with the synthesis of ATP by the Rnf complex, which is essential for its energy metabolism during autotrophic growth[5–8].

[1]SYNMIKRO Research Center and Department of Chemistry, Philipps-University of Marburg, Marburg, Germany. [2]Department of Molecular Microbiology & Bioenergetics, Institute of Molecular Biosciences, Johann Wolfgang Goethe University, Frankfurt am Main, Germany. [3]Department of Biochemistry and Biophysics, Stockholm University, Stockholm, Sweden. [4]Helmholtz Munich Cryo-Electron Microscopy Platform, Helmholtz Munich, Neuherberg, Germany. [5]These authors contributed equally: Anuj Kumar, Jennifer Roth, Hyunho Kim, Patricia Saura. ✉e-mail: ville.kaila@dbb.su.se; jan.schuller@synmikro.uni-marburg.de; vmueller@bio.uni-frankfurt.de

The Rnf complex catalyses the oxidation of low-potential ferredoxin (Fd, $E_0' = -430$ to $-500$ mV), generated by the electron-bifurcating hydrogenase (HydABC)[9], to reduce nicotinamide adenine dinucleotide (NAD$^+$) ($E_0' = -320$ mV), transducing the redox energy into an electrochemical sodium ion gradient (or sodium motive force, SMF) across the membrane that drives the synthesis of ATP. When operating in reverse mode, the Rnf complex harnesses the energy from the SMF to catalyse the endergonic NADH-driven reduction of Fd, required for the chemically challenging reduction of nitrogen gas (N$_2$)[10]. In *A. woodii*, Rnf functions in its reverse mode during growth on low-energy substrates (lactate, ethanol, methanol) to balance the redox pools and provide reduced ferredoxin for CO$_2$ reduction in the Wood-Ljungdahl pathway, and for nitrogen fixation[7]. Despite these functional insights, its redox-driven Na$^+$ translocation mechanism remains elusive and highly debated[11–14].

Thermodynamically, the Rnf complex operates at rather low redox potentials, with Fd as an electron donor ($E_0' \approx -450$ mV) and NAD$^+$ as an electron acceptor ($E_0' = -320$ mV), whilst NADH is re-oxidised by CO$_2$ reduction to acetate. The free energy ($\Delta G_0'$) of the Rnf-catalysed reaction is relatively small, *ca.* $-6$ kcal mol$^{-1}$ ($-25$ kJ mol$^{-1}$/$-260$ mV for 2 e$^-$; $-130$ mV per electron), which can be used to pump up to two Na$^+$ across the cytoplasmic membrane at a SMF of $<130$ mV. In contrast to other energy-transducing ion pumps that often operate with much larger driving forces, the energetically constrained redox-driven sodium ion pumping in Rnf may pose structural and mechanistic boundaries to establish effective ion transport without significant dissipation of energy, as also supported by the reversibility of the pumping machinery.

The Rnf complex thus represents a unique biochemical system adopted by an evolutionarily divergent family of enzymes. Rnf is an ancestor of the Na$^+$-pumping NADH:ubiquinone oxidoreductase (Nqr)[15], which also generates a SMF, but employs NADH as the electron donor and ubiquinone ($E_0' = 90$ mV) as an electron acceptor, thus operating at a substantially more positive redox potential range and larger driving force ($\Delta G_0' = -18$ kcal mol$^{-1}$/$-79$ kJ mol$^{-1}$). To this end, the membrane-embedded subunits between Rnf and Nqr (RnfA/E/D/G and NqrE/D/B/C) are conserved, and suggest that the complexes may employ a similar Na$^+$ translocation mechanism. To operate with NADH/quinone instead of Fd/NAD$^+$ as electron donor/acceptors, the Nqr complex incorporated the NqrF subunit, which is related to NAD(P)H ferredoxin reductase (FNR)[16,17] and oxidises NADH by FAD. Moreover, the soluble NqrA subunit (equivalent to RnfC) lost its redox cofactors, while the quinone binding takes place in NqrB (equivalent to RnfD). Notably, the Rnf and Nqr complexes are the primary respiratory enzymes for a large group of pathogenic bacteria[18,19], characterised by unique mechanisms and structural motifs absent in human proteins. This distinctiveness presents an opportunity for pharmacological exploitation, potentially paving the way for the development of innovative antibiotics. However, despite extensive structural studies of both Rnf[12] and Nqr[11,13], the mechanism by which the electron transfer is coupled to sodium ion pumping remains elusive and highly debated, with several conflicting suggestions on the location of the ion translocation pathways[11–14].

Here we integrate cryo-electron microscopic (cryo-EM) studies, with biochemical functional assays, and atomistic molecular dynamics simulations to probe the mechanism underlying the redox-driven sodium ion translocation in the Rnf complex from *A. woodii*. Our combined results show that Rnf operates by an alternating-access mechanism that drives the Na$^+$ translocation by redox-mediated conformational changes of a membrane-embedded iron-sulphur (FeS) cluster, thus combining distinct mechanistic features of both primary active transporters and redox-driven ion pumps.

## Results

### Purification of the Rnf complex from *A. woodii*

The Rnf complex was produced using a plasmid-based production system in *A. woodii* that allows it to overexpress and purify an affinity-tagged version of the complex from its native host, as described previously (Supplementary Fig. 1)[8]. The presence of flavin cofactors in the complex has been previously reported [8,20], whereas we could identify flavin mononucleotide (FMN) in RnfC, RnfD, and RnfG and a riboflavin (RBF) in RnfD based on the structural data (see below). Moreover, the iron (Fe) content determined using calorimetric methods revealed that the purified complex contained $41.8 \pm 1.5$ mol iron/mol Rnf (Supplementary Table 2). The purified Rnf complex was active, and catalysed the ferredoxin-dependent reduction of NAD$^+$ with an activity of $7.1 \pm 1.1$ U mg$^{-1}$ (Supplementary Fig. 1, Supplementary Table 1).

### The molecular architecture of the Rnf complex

To reveal the structural basis of how the Fd:NAD$^+$ oxidoreductase activity couples to the sodium ion translocation, we subjected the complex to redox-controlled cryo-EM single particle analysis. As Rnf is highly sensitive to oxygen, we prepared the cryo-EM grids under strictly anoxic conditions in an anaerobic chamber filled with 95% N$_2$ and 5% H$_2$ to ensure the structural integrity of the complex. We obtained two structural snapshots of the Rnf complex that represent states along the reverse and direct electron transfer directions during catalytic turnover conditions: (1) an NADH-reduced state resolved at 3.3 Å resolution, and (2) a Fd-reduced state at 2.8 Å resolution, with the Fd purified from *Clostridium pasteurianum* and reduced by catalytic amounts of a CO dehydrogenase obtained from *A. woodii* (Supplementary Figs. 2–6, Supplementary Table 4). We also determined the structure of the Rnf complex in its *apo* state with an overall resolution of 3.0 Å (Supplementary Fig. 4). Taken together, the structures provide insight into both the endergonic (NADH→Fd) and exergonic (Fd→NADH) electron transfer directions.

Since all determined structures have overall the same organisation and cofactor content, we focused our description on the Rnf complex with the bound NADH (Fig. 1, Supplementary Fig. 2, Supplementary Fig. 6, Supplementary Movie 1). The complex comprises six subunits, with three integral membrane subunits RnfA/D/E, two membrane-anchored proteins RnfB and RnfG, and an attached cytoplasmic protein, RnfC. On the cytoplasmic side, the complex comprises RnfC and RnfB that are responsible for the interactions with the soluble electron carriers, whilst the periplasmic side harbours the flavoprotein RnfG. The NADH-binding RnfC subunit is the only subunit that is not membrane-bound. RnfC has a globular flavin-binding domain and a small helical C-terminal domain that binds two [4Fe4S] clusters (C1 and C2), with similarities to the FMN site of respiratory Complex I (Nqo1) and HydABC (HydB)[21]. In the determined structure, we observed a well-resolved density for a bound NADH molecule, which forms stacking interactions to the non-covalently bound FMN cofactor (Supplementary Fig. 5d). RnfC binds to the poly-ferredoxin subunit RnfB on the cytoplasmic side of the complex, with the interaction mediated by a short C-terminal helix of RnfC. In addition, unstructured N- and C-terminal loops of RnfD bind to the cavities within RnfC that stabilise the subunit (Supplementary Fig. 5c) and could be partially resolved in our cryo-EM data. In RnfC, all cofactors are linearly arranged to the polytopic membrane protein RnfD and located at distances that could enable an efficient electron transfer across the complex (Fig. 1c, d).

RnfD comprises ten transmembrane (TM) helices that bind two flavin cofactors and extend towards the centre of the membrane plane, thereby directly routing the electrons across the membrane to the periplasmic RnfG subunit (Fig. 1). In the pocket connected to the cytoplasmic side, RnfD binds the redox cofactor riboflavin (RBF), similar to other known Rnf and Nqr structures[12,14]. RBF forms hydrogen-bonds through its ribityl moiety with several conserved

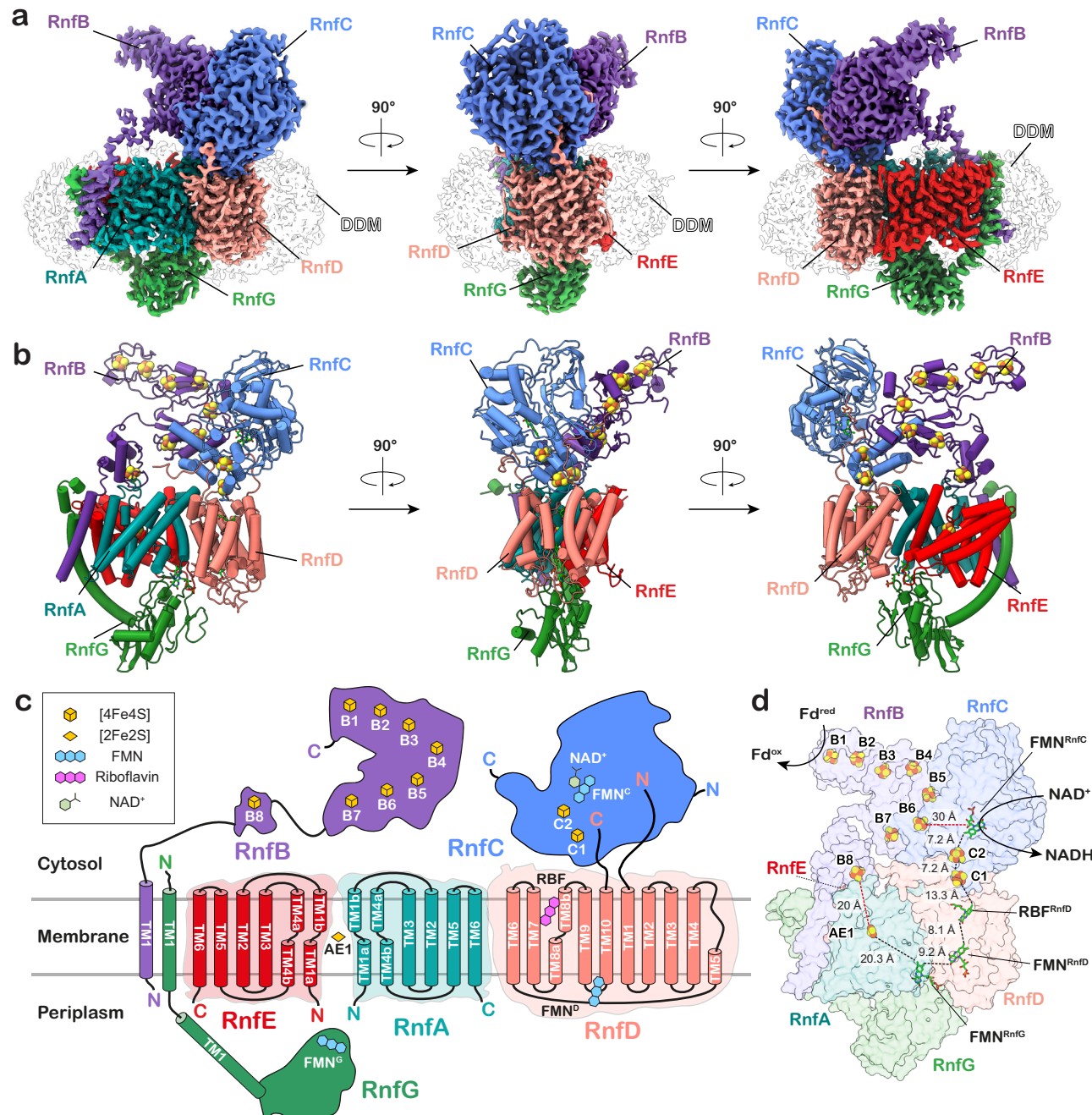

**Fig. 1 | Molecular architecture of the Rnf complex. a** Three-dimensional segmented cryo-EM density map of the Fd-reduced Rnf complex coloured by subunits. **b** Overall structure of the Rnf complex shown in cartoon representation. **c** Graphical representation of the Rnf complex showing the topology of the membrane-bound area along with the cofactors harboured by each subunit. **d** Cofactor content in the Fd-reduced Rnf complex, with *edge-to-edge* distances between the FeS clusters and cofactors shown in ångströms. See Supplementary Fig. 6 for a comparison of all resolved cryo-EM structures.

residues (D249$^{RnfD}$, N123$^{RnfD}$), which could stabilise the semiquinone form of the cofactor (Fig. 2d). Additionally, RnfD contains an FMN$^D$ cofactor covalently linked to T156$^{RnfD}$, located in a pocket connected to the periplasmic side that could mediate the electron transfer from FMN$^G$ to RBF, or vice versa during reverse electron transfer (Fig. 1d). FMN$^D$ is in proximity to the mobile hydrophilic domain of the RnfG subunit. The periplasmic RnfG subunit is anchored to the membrane via a long stalk helix (TM1$^G$), which connects to a mobile globular domain that covalently binds FMN$^G$ via T185$^{RnfG}$ in its antiparallel βαβ-fold (Figs. 1c, 2c). RnfG also forms a major contact interface at the centre of the cavity enclosed by the RnfA/E subunits, with only minor contacts with RnfD.

RnfA and RnfE are homologous subunits with a sequence identity of 29% (Supplementary Fig. 7b), forming a dimer with a C2 pseudo-symmetric axis parallel to the membrane plane. In the cryo-EM map, we observed at the pseudo-symmetric axis a rhombus-shaped density resembling a [2Fe2S] cluster (Fig. 2b). The density extends across two pairs of conserved cysteine residues from each subunit (C25$^{RnfA}$, C113$^{RnfA}$, C25$^{RnfE}$, C108$^{RnfE}$) that ligate the [2Fe2S] cluster (Fig. 2b). We note that the presence of such [2Fe2S] cluster (from here on the "AE1 cluster") buried within the membrane is highly unusual and found only in Rnf and the related Nqr complex[13,14].

RnfB serves as the site for Fd binding that is anchored to the membrane via its long transmembrane helix (TM1$^B$), which interacts

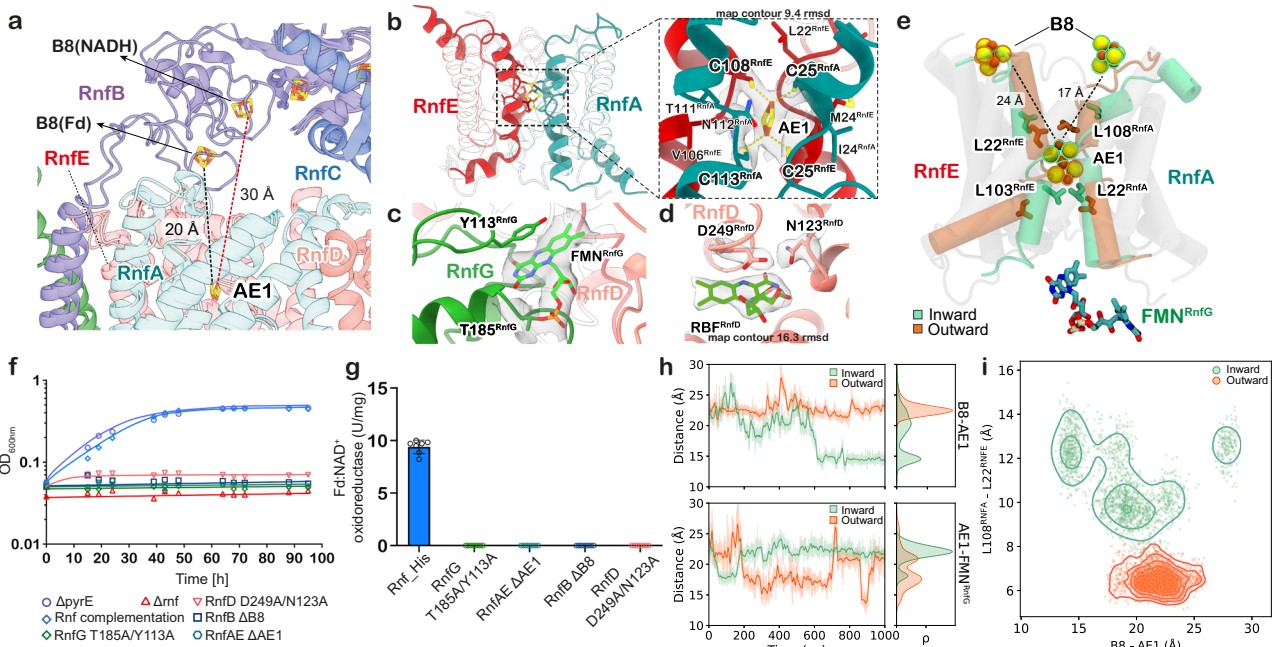

**Fig. 2 | Structural and functional characterisation of the electron transfer pathway in the Rnf complex. a** The B8 domain in RnfB shows different conformations in the NADH-reduced and Fd-reduced cryo-EM structures. **b–d** Segmented cryo-EM density encasing AE1, FMN$^{RnfG}$, and RBF, highlighting the residues coordinating the cofactors. The cysteine residues coordinating the AE1 cluster in (**b**) are shown, with the map contour RMSD level indicated. In (**d**), the hydrogen-bond distances between D249/N123 and RBF are indicated in ångströms, including the map contour level. **e** Superimposed structures of RnfA/E from MD simulations classified in the inward (green) and outward (orange) conformations. Conserved hydrophobic residues on TM4a (L103$^{RnfE}$, L108$^{RnfA}$) gate the access to the AE1 cluster from either the intracellular or extracellular bulk, depending on the alternate-access state. In the inward conformation, the distance between the B8 and AE1

clusters is reduced to *ca.* 17 Å as compared to 24 Å in the outward conformation. **f, g** Rnf complementation strain rescues the growth of *A. woodii* on $H_2$ and $CO_2$. The Rnf strains containing mutation of residues that stabilise the cofactors could not recover growth on $H_2$ and $CO_2$ or show any Fd:NAD$^+$ oxidoreductase activity. Activities are the mean ± s.e.m. of three independent biological replicates, measured in triplicates ($n = 3$). **h** *Edge-to-edge* distances of cofactors B8-AE1 (top), and AE1-FMN$^G$ (bottom) from MD simulations initiated from Fd-reduced Rnf structure. Shown are both the raw data (transparent line) and the rolling average (solid line). **i** Analysis from MD simulations showing the distance correlation between the conserved gating residues L108$^{RnfA}$ and L22$^{RnfE}$, and the B8 and AE1 clusters. The MDs are clustered by the inward (green) or outward (orange) conformation.

with TM1$^G$ of RnfG (Fig. 1). TM1$^B$ is connected to a highly dynamic Fd-like domain that contains a single [4Fe4S] cluster (B8-domain), and also links to the cytosolic part of RnfB, harbouring seven [4Fe4S] clusters (B1-B7). Although the core of RnfB shows low flexibility, its C-terminal region, as well as the B8-domain, are highly mobile, as indicated by the poor density and low local resolution (Supplementary Figs. 2f, 4f, and 5a, f, g). We further note that the C-terminal region of RnfB is decorated with positively charged lysine and arginine residues that could form a dynamic "fishing hook" for the electron transfer protein ferredoxin (Supplementary Fig. 8g).

In summary, our high-resolution structure of the Rnf complex could resolve all the structural features and the cofactor content (Fig. 1a, b, d and Supplementary Fig. 5). In comparison to previous Rnf structures, the Rnf complex from *A. woodii* contains the largest RnfB subunit, which harbours eight [4Fe4S] clusters, in contrast to other isoforms that contain three to six [4Fe4S] clusters[12,22]. Moreover, the structure of the Rnf complex from *Clostridium tetanomorphum* was poorly resolved, and the resolution was insufficient to identify the membrane-bound [2Fe2S] cluster[12]. In our structure, we could clearly resolve the [2Fe2S] cluster bound in the membrane, in complete agreement with other well resolved cryo-EM structures of Nqr and Rnf complexes[13,14,22]. Additionally, the recent Rnf1 structure from *Azotobacter vinelandii*[22] has only a partially resolved RnfB.

**Redox-driven conformational changes gate electron transfer across the Rnf complex**

As shown by our functional assays (Supplementary Fig. 1c), the Rnf complex transfers electrons from the reduced Fd to NAD$^+$.

Additionally, previous studies have shown that the redox reaction is coupled to Na$^+$ pumping across the membrane[8]. The cytosolic RnfB and RnfC subunits sit next to each other and are responsible for binding the electron carriers Fd and NAD$^+$, respectively (Fig. 1c, d). Fd is likely to dock near the B1-B2 cluster region in RnfB (Supplementary Fig. 8g), which has structural similarities to HydABC[21]. From RnfB, the electron is transferred along the FeS chain (B3-B7) to the terminal B8 cluster (Figs. 1d, 2a), whilst no direct electron transfer is likely to occur between RnfC and RnfB subunits due to the large distance (>30 Å) between the closest FeS centres (Fig. 1d). Instead, our resolved structures suggest that the electrons transfer through the membrane domain of the complex (RnfA/E and RnfD) and reach the soluble electron carriers via the periplasmic RnfG in an overall U-shaped pathway (Fig. 1d). While previous studies also proposed similar electron transfer pathways and coupled domain rearrangements[12–14], experimental evidence of the conformational changes linked to the sodium ion translocation pathways remain elusive. The cryo-EM experiments conducted by reducing the Rnf complex with either Fd or NADH, enabled us to investigate the forward exergonic and reverse endergonic electron transfer modes, and to shed light on the redox-dependent conformational dynamics of the complex that drives the sodium pump.

Remarkably, in the Fd-reduced state of the complex, the Fd-like domain becomes highly dynamic relative to the NADH-treated sample, with sub-classification allowing us to identify distinct intermediates (Supplementary Fig. 3). This alternate conformation brings the B8 cluster to a closer proximity (20 Å) of the AE1 cluster (Fig. 2a), which is located within the membrane region of the protein (see below). The

structure of the Rnf complex with the B8 cluster closer to the membrane shows a possible intermediate state, induced by the reduction from Fd that has not been determined in previous studies. In addition, 3D masked classification on RnfG revealed a state where the subunit and its cofactor FMN$^G$ showed a minor movement, slightly reducing the distance to the AE1 cluster (Supplementary Fig. 3c), but also suggesting that the conformational space is hindered by the tight binding of the detergent belt (see Methods). A similar effect was previously reported for the NqrC subunit in the crystal structures of the Nqr complex (but see below).

To probe how the different states are linked to the electron transfer process, and to elucidate how these could drive the sodium ion translocation, we performed large-scale atomistic molecular dynamics (MD) simulations using our experimentally determined cryo-EM structures as starting points (*see Methods*, see Supplementary Table 6, Supplementary Fig. 8a–f), but embedded in a model of a biological membrane. The techniques are highly complementary, but the MD simulations allow us to further probe the dynamics of transient redox states during the catalytic cycle that are difficult to experimentally isolate. The overall dynamics of the Rnf complex inferred from MD simulations resembled the local resolution of the cryo-EM structures, showing higher dynamics in the peripheral subunits as compared with the membrane-bound regions (Supplementary Fig. 5g), but also revealed distinct conformational changes linked to the sodium ion translocation.

Remarkably, the dimeric RnfA/E interface relaxes into two distinct conformations during the MD simulations, with important implications for the Na$^+$ translocation mechanism. In simulations initiated from the NADH-reduced structure, the RnfA/E dimer samples an extracellular/periplasmic open (outward-facing) conformation (Fig. 2e). In stark contrast, the Fd-reduced state relaxes into conformations where RnfA/E has membrane access either from the outward-facing or from the cytosolic open, inward-facing states (Fig. 2e, Supplementary Fig. 9, Supplementary Table 6), but no direct access across the membrane, which is central for avoiding ion leaks. During the conformational changes, the B8-AE1 distance decreased from *ca.* 20 Å (from the cryo-EM structure) further to 15-17 Å in our MD simulations (Fig. 2e, h, Supplementary Fig. 10a, c), bringing the two cofactors into feasible electron transfer distances in the inward conformation. These conformational changes are enabled by the insertion of the B8-domain (residues 30-100 of RnfB, see Supplementary Fig. 8e, f) between the RnfA/E subunits together with conformational changes in TM1/4 of RnfA. More specifically, helix TM4a, moves towards/away from the perpendicular helix TM1a, thus gating the access of ions and water molecules from the bulk solvent to the protein interior (Fig. 2e, Supplementary Fig. 9a–d), whilst the conserved hydrophobic residues, L22$^{RnfA}$ and L103$^{RnfE}$ on the extracellular side, and L22$^{RnfE}$ and L108$^{RnfA}$ on the intracellular side of TM1a and TM4a, establish an occluded access during the conformational switching (Fig. 2e, i, Supplementary Figs. 9a, b, 10). The identified residues are highly conserved in Rnf and Nqr complexes, further supporting their functional relevance in Na$^+$-pumping (Supplementary Fig. 7b, see below).

Importantly, reduction of the AE1 cluster by B8 in the inward-open state leads to sodium ion binding near the AE1 centre (Fig. 3e–h, Supplementary Fig. 12). A conserved lysine residue (K32) together with other nearby charged residues (R154, E158) of RnfA, interact with the B8 cluster and could aid in sensing its redox changes (Supplementary Fig. 10a, b and see below).

Following the conformational changes during the inward/outward-switching, the electron is subsequently transferred from AE1 to FMN$^G$, on the other side of the RnfA/E dimer. The *edge-to-edge* distance between the AE1-FMN$^G$ is >20 Å in our cryo-EM structure of the Fd-reduced state (Fig. 1d, Supplementary Fig. 6), whilst in the MD simulations this distance decreases to around 16 Å by conformational

changes in RnfG (Fig. 2e, h, Supplementary Fig. 10d, e). Remarkably, we observe that the conformational switching of FMN$^G$ is correlated with the inward/outward transition of RnfA/E, which could control the rate of electron transfer by conformational gating (Supplementary Fig. 10d, e, Supplementary Table 8). From FMN$^G$, the electron is transferred to the flavin cofactors in RnfD (FMN$^D$ and RBF, Fig. 1d, Supplementary Fig. 10a, c), and further to the cytosolic side of RnfC, where NAD$^+$ is reduced by FMN$^C$, thus completing the proposed U-shaped electron transfer pathway (Fig. 1d). When accounting for the observed conformational changes during our MD simulations, our kinetic simulations predict that the electron is transferred across the complex with an overall rate of milliseconds (Fig. 4c, Supplementary Table 8). The proposed pumping model (Fig. 4a) also operates efficiently at 180 mV SMF (Fig. 4b, c, Supplementary Fig. 13).

To validate our proposed electron transfer pathway from RnfB to RnfC, we created site-specific mutations to remove the key redox cofactors along this pathway, and measured the oxidoreductase activities in the purified variants. We also introduced the mutated genes into the *rnf*-negative strain of *A. woodii* and studied the mutants for autotrophic growth. Mutation of the cysteines coordinating the B8 [4Fe4S], the AE1 [2Fe2S] clusters (Figs. 1d, 2a, b), or the residues coordinating FMN$^G$ (T185A, Y113A) in RnfG and RBF (N123A, D249A) in RnfD subunit (Fig. 2c, d), lead to strains unable to grow (Fig. 2f), whilst the purified Rnf variants did not catalyse Fd-dependent NAD$^+$ reduction (Fig. 2g, Supplementary Fig. 14a), supporting the functional relevance of these redox cofactors. Analysis of the iron content revealed that variants lacked two mol of iron for ΔAE1 and four mol of iron for ΔB8 variants as compared to the wildtype Rnf, whereas for the ΔFMN$^G$ and ΔRBF$^D$ the iron content was similar to the wildtype (Supplementary Table 2).

## Redox state of B8/AE1 controls sodium ion binding in RnfA/E

Our MD simulations reveal that Na$^+$ enter RnfA/E from the intracellular side only in the inward-facing conformation, and from the extracellular side in the outward-facing conformation (Fig. 3g, h, Supplementary Fig. 12a, b, Supplementary Movie 2), with the "buried" binding-site (see below) accessible for the Na$^+$ only upon the reduction of the AE1 cluster, suggesting that the Na$^+$ pumping is coupled to the reduction of the cluster (Fig. 3e–h, Supplementary Figs. 12b–d, 15). Interestingly, our cryo-EM structure of Fd-reduced Rnf also reveals density in the well-resolved core of the RnfA/E interface that could arise from Na$^+$ in the narrow cavity around the [2Fe2S] cluster region (Fig. 3a, Supplementary Fig. 12e, f), further supporting the MD data. Taken together, this suggests that the sodium ions interact initially with residues close to the cytosolic surface (E115$^{RnfE}$, P107$^{RnfA}$ backbone, T110$^{RnfA}$), from which they can reach a buried Na$^+$ binding site near the AE1 cluster (Fig. 3a, b, e–h, Supplementary Fig. 12b–d), whilst Y105$^{RnfE}$ could function as a lid that undergoes conformational changes upon inward/outward transition (Supplementary Fig. 7c–e). The Na$^+$ binding at the AE1 cluster is supported by the backbone of V106$^{RnfE}$ and T111$^{RnfA}$, as well as transient interactions with the hydroxyl group of T110$^{RnfA}$ (Fig. 3e–h, Supplementary Fig. 12b–d). The inward/outward transition, induced by conformational changes in TM4 of RnfA, disrupts an ion-paired network comprising E115$^{RnfE}$, R67$^{RnfE}$, and E88$^{RnfA}$ (Fig. 3b, Supplementary Fig. 9c, d). The conformation of this ion-pair modulates the binding affinity of Na$^+$, with stable ion binding observed in the open ion pair conformation (Fig. 3b, Supplementary Fig. 9c). Our cryo-EM structure of the Fd-reduced Rnf complex indeed supports the two distinct conformations of R67$^{RnfE}$, with a Na$^+$ resolved near E115$^{RnfE}$ in the open ion-pair conformation (Fig. 3b).

As mentioned above, the Na$^+$ affinity for the buried inward-binding site is highly sensitive to the redox state of the AE1 cluster. In the oxidised state of AE1, the Na$^+$ binding is weakly exergonic (*ca.* −1 kcal mol$^{-1}$), but increases (up to −3 kcal mol$^{-1}$) upon reduction of AE1 (Supplementary Fig. 15), suggesting that the electron transfer from B8

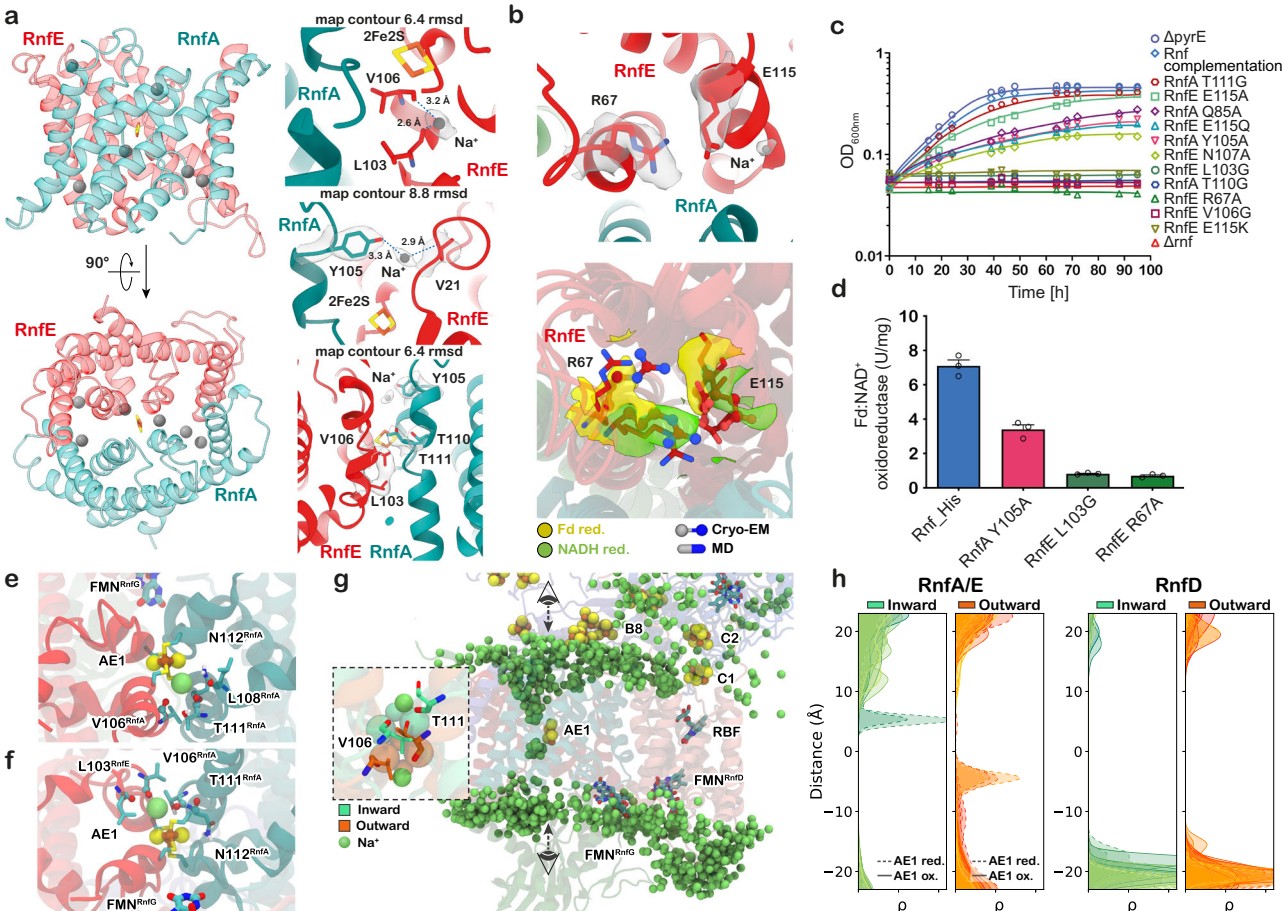

**Fig. 3 | RnfA/E subunits pump the sodium ion across the membrane gated by AE1 [2Fe2S] cluster. a** Left: RnfA/E dimer containing the AE1 cluster and bound Na$^+$ (in grey), in the Fd-reduced cryo-EM structure. Right: Close-up of the Na$^+$ binding sites in the RnfA/E subunits around the AE1 cluster, with the map contour RMSD level indicated. **b** Top: Cryo-EM structure of the R67$^{RnfE}$-E115$^{RnfE}$ ion pair with Na$^+$ bound close to the glutamate residue with the densities shown in grey transparent surface. Bottom: Comparison of the R67$^{RnfE}$-E115$^{RnfE}$ ion pair conformation between different states from both cryo-EM and MD simulation structures shows similarities. **c, d** Rnf complementation strain rescues the growth of *A. woodii* on H$_2$ and CO$_2$. The Rnf strains containing mutation of residues involved in Na$^+$ binding and translocation showed either reduced growth or no phenotype. Mutation of residues involved in Na$^+$ pumping or gating (R67, L103), showed drastically reduced Fd:NAD$^+$ oxidoreductase activity as compared to the wildtype protein. Activities are the mean ± s.e.m. of three independent biological replicates, measured in triplicates (*n* = 3). Structural snapshots from MD simulations (S15, S16), showing sodium ion binding in the intracellular (**e**) and extracellular (**f**) buried binding sites within RnfA/E (green/red) near the AE1 cluster. The binding is stabilised by backbones of V106$^{RnfE}$, L103$^{RnfE}$, and T111$^{RnfA}$, as well as by N112$^{RnfA}$ side chain in the intracellular side. The perspective for the MD snapshots is indicated in (**g**). **g** Dynamic ensemble of Na$^+$ (lime-green spheres) in the membrane domain (RnfA, RnfE, RnfD) from two different MD simulations (Inward: S15, Outward: S16; see Supplementary Table 6). Inset: The sodium ions interact with the backbone carbonyls of the T111$^{RnfA}$ and V106$^{RnfE}$ in the inward (green) and outward (orange) conformations. **h** Probability density of Na$^+$ from the RnfA/E (left) and RnfD (right) centres during the MD simulations. The centre of the distribution is set to the AE1 cluster for RnfA/E and M246$^{RnfD}$ for RnfD. All MD simulations clustered into inward (green) / outward (orange) conformations were included in the analysis.

to AE1 initiates the Na$^+$ translocation by pulling in cytosolic Na$^+$ via electrostatic effects. Similarly, the MD simulations of the outward-facing conformations reveal Na$^+$ binding at the extracellular side, with similar redox-state dependence on the AE1 cluster as in the inward-state (Fig. 3e–h, Supplementary Fig. 15). In the outward-state, the Na$^+$ transiently interact with negatively charged and polar residues on the extracellular surfaces of the subunits RnfA (D64$^{RnfA}$, T73$^{RnfA}$, E126$^{RnfA}$), RnfE, and RnfG (E116$^{RnfG}$, E86$^{RnfG}$), before ultimately binding to the buried Na$^+$ binding site that is stabilised by the backbones of V106$^{RnfE}$, T111$^{RnfA}$, and L103$^{RnfE}$ (Fig. 3f, Supplementary Fig. 12b–d). Interestingly, two of the backbone carbonyls (V106$^{RnfE}$ and T111$^{RnfA}$) can bind Na$^+$ either from extracellular side (in the outward-conformation) or the intracellular side (in the inward-conformation). These residues are located on flexible loops between the helices TM4a and TM4b, and can adapt their conformation to coordinate Na$^+$ from either side (Fig. 3g, Supplementary Fig. 9e, f). This suggests that the Na$^+$ transport across the RnfA/E subunit is facilitated by the carbonyl moieties of V106$^{RnfE}$

and T111$^{RnfA}$, transitioning via a putative "occluded" state, where Na$^+$ is shielded from both the extracellular and the intracellular side.

The inward-facing conformation of RnfA/E enables the movement of the B8-domain deeper into the membrane region, as well as the shifting of the AE1 cluster towards the cytoplasmic side of the membrane (Supplementary Fig. 9e, g, Supplementary Movie 3), which brings the B8 and AE1 clusters into electron transfer distance (*ca.* 15–17 Å, see Supplementary Fig. 10c). In contrast, the occluded RnfA/E interface at the cytosolic surface observed in the outward-facing conformation, prevents the flexible B8 domain from approaching the AE1 centre (see above, Figs. 1c, 2e, h, i, Supplementary Fig. 10c).

To estimate the energetics of the inward-outward transition, we performed free energy calculations using a variant of the string simulation method (see Supplementary Methods, Supplementary Fig. 19a–k, and Supplementary Movies 6 and 7). Our free energy calculations suggest that during the conformational transition, a sodium ion follows a continuous pathway from the cytosolic to the

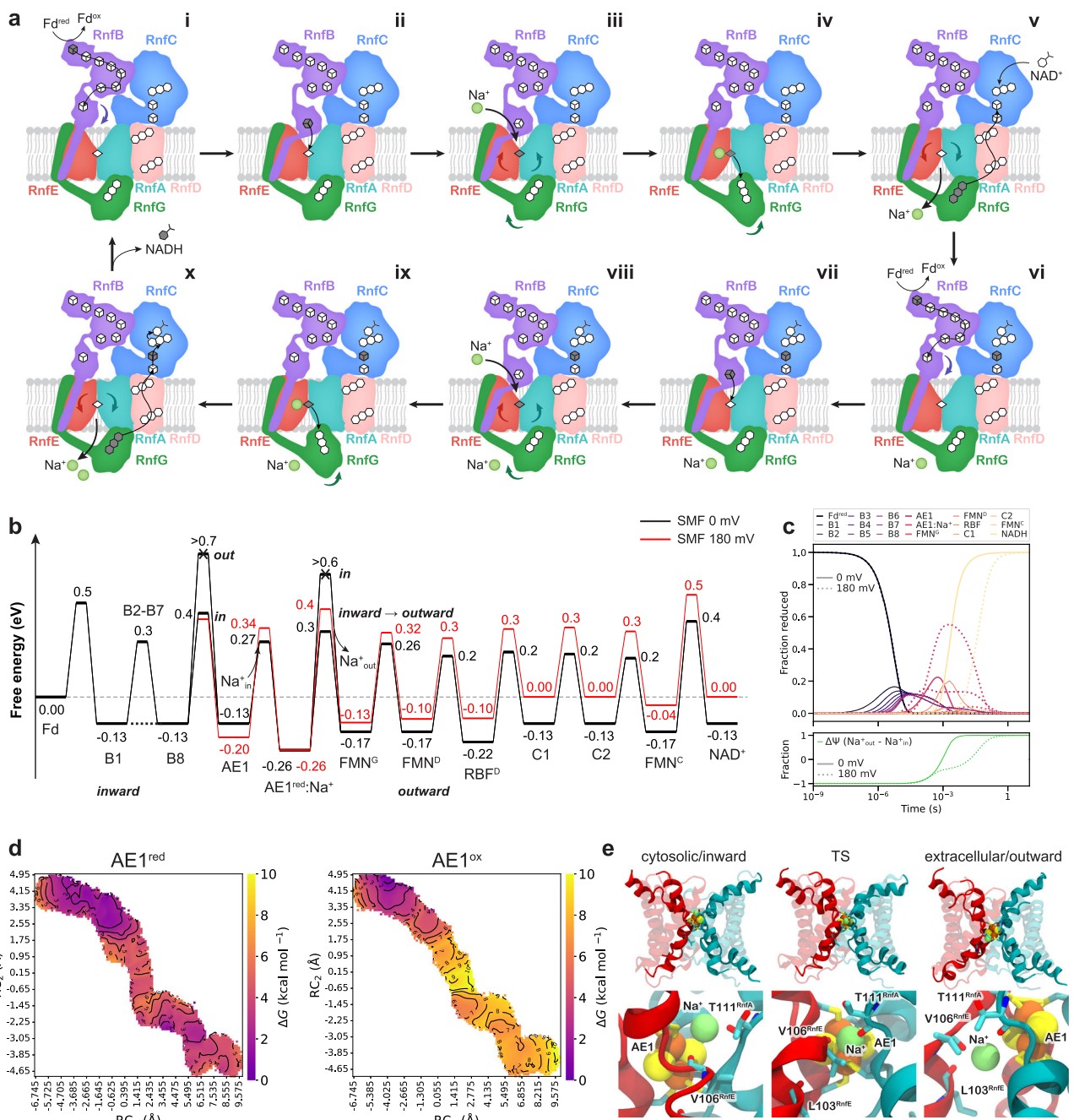

**Fig. 4 | Redox-driven Na⁺ translocation mechanism in the Rnf complex.**
**a** Schematic representation of the redox-driven Na⁺ translocation mechanism. **i**: Fd transfers an electron via a chain of FeS cluster in RnfB to the terminal B8 cluster. **ii**: Reduction of B8 induces a conformational change in the B8-domain, which approaches RnfA/E on the cytosolic side. The conformational changes result in a shorter distance between B8 and AE1, enabling the electron transfer. **iii**: Reduction of AE1 increases the Na⁺ binding affinity in RnfA/E that leads to Na⁺ uptake from the cytosol. **iv**: After Na⁺ binding next to AE1, RnfA/E undergoes an inward/outward transition that, in turn, moves RnfG closer to RnfA/E. **v**: Electron transfer from AE1 to FMN^G lowers the binding affinity for Na⁺, and results in the Na⁺ release to the extracellular side. The electron is transferred from FMN^G to the RnfC FeS via the flavins on RnfD, in a process that couples to the relaxation of RnfA/E to its inward-open state. The reduction of C2 increases the affinity of RnfC for NAD⁺, which binds

next to the FMN^C. **vi-ix**: similar to previous steps, another electron transfer from Fd couples to the translocation of a second Na⁺. **x**: Two-electron reduction yields FMNH⁻, which reduces NAD⁺ to NADH via a hydride transfer, initiating a new pumping cycle. **b** Schematic free energy profile of the proposed Na⁺ pumping mechanism (see Supplementary Methods, and Supplementary Table 8). **c** Kinetic simulations of the redox-driven Na⁺ translocation process with a 0 mV (solid line) and 180 mV (dotted line) SMF. **d** 2D free energy profiles of the inward-outward transition coupled to the Na⁺ translocation with AE1 reduced (*left*) or oxidised (*right*). RC₁−distance between TM4 of RnfA and RnfE; RC₂−distance of Na⁺ and the centre of mass of RnfA/E, projected onto the Z-axis (see Supplementary Figs. 18−20 for details). **e** Structures along the free energy profile with AE1 reduced: RnfA/E (*top*), closeup of the Na⁺, AE1, and residues interacting with Na⁺ (*bottom*). *Left*: inward state; *middle*: transition state (TS); *right*: outward state.

extracellular site (Supplementary Fig. 20a), in a process where the ion coordinates to the carbonyl group of the T111$^{RnfA}$ and V106$^{RnfE}$, backbone, next to the AE1 cluster (Fig. 4d, e, Supplementary Fig. 20a). The motion of the TM4a helix couples to the opening of the E115$^{RnfE}$-R67$^{RnfE}$-E88$^{RnfA}$ ion-pair network (Supplementary Fig. 20b–d and see Supplementary Fig. 9c) and the rotation of the carbonyl groups (Supplementary Fig. 20e–g), which could assist in pulling the Na$^+$ from the *inward* to *outward* vestibules within the buried binding site (Fig. 4d, e, Supplementary Fig. 20a). The 2D free energy profiles suggest that the free energy barrier is rather small ($\Delta G^{\ddagger} = +4$ kcal mol$^{-1}$) and the process is nearly isoenergetic when the AE1 cluster is reduced (Fig. 4d). However, in stark contrast, when the AE1 cluster is oxidised, the process becomes endergonic ($\Delta G^{\ddagger} = +8$ kcal mol$^{-1}$), and the reaction has a significantly higher free energy barrier ($\Delta G^{\ddagger} = +10$ kcal mol$^{-1}$), thus supporting that reduction of AE1 is coupled to the sodium ion translocation (Fig. 4d, Supplementary Fig. 19e, h). The redox state dependence is also supported by solvation free energy calculations of the Na$^+$ along the conduction pathway, suggesting that reduction of AE1 stabilises the Na$^+$ binding by 1-4 kcal mol$^{-1}$ along the pathway (Supplementary Fig. 15f). These results support our general mechanistic proposals of the redox-driven conformational switching coupled to ion transport, but we note that future work is required to fully characterise the complete free energy landscape and how this process drives the ion transport across the membrane.

To experimentally validate the Na$^+$ translocation pathway, we performed site-directed mutagenesis on the proposed Na$^+$ binding residues. To this end, we studied whether the mutants were able to grow on H$_2$ and CO$_2$ in the presence of 20 and 2 mM NaCl, as well as by determining the Fd:NAD$^+$ oxidoreductase activity of the purified variants. We first probed the cytosolic Na$^+$ binding site by introducing R67A, E115A, E115Q, and E115K substitutions in RnfE and analysing autotrophic growth at 20 mM NaCl. The R67A mutant did not grow on H$_2$ and CO$_2$ (Fig. 3c, Supplementary Table 3). Similarly, autotrophic growth was also drastically reduced in the E115Q mutant, and completely abolished in the E115K mutant, although the E115A mutant grew similar to the wildtype (Fig. 3c, Supplementary Table 3). These data suggest that the R67-E115$^{RnfE}$ ion pair plays a central role in the initial binding of Na$^+$ and in modulating the conformational dynamics of the RnfA/E interface, as also suggested by our free energy calculations. Next, we introduced the N107A, L103G, and V106G mutations in RnfE, in the vicinity of the AE1 cluster. The N107A strain could partially recover autotrophic growth, whereas growth was completely abolished in L103G and V106G mutants, consistent with our MD simulations and free energy calculations, and cryo-EM data (Fig. 3c, Supplementary Table 3). In RnfA, we further probed the effect of Y105A, Q85A, T111G, and T110G mutations. The Y105A mutant showed reduced growth rates, while T110G did not grow. T111G showed no phenotype, implying that at least the hydroxyl group of the sidechain is not significant for Na$^+$ translocation (Fig. 3c, Supplementary Table 3). The purified Rnf complexes harbouring R67A, Y105A, and L103G mutations, could catalyse only a 10% ($0.7 \pm 0.2$ U mg$^{-1}$), 50% ($3.4 \pm 0.7$ U mg$^{-1}$), and 11% ($0.8 \pm 0.2$ U mg$^{-1}$) oxidoreductase activity, respectively, as compared to the wildtype ($7.1 \pm 0.1$ U mg$^{-1}$) (Fig. 3d, Supplementary Fig. 14c, d Supplementary Table 1). Moreover, the strains harbouring mutations in the RnfE subunit (see above), as well as Y105A and T111G variants in RnfA, could not grow under sodium-depleted conditions (under 2 mM NaCl) (Supplementary Fig. 14e), thus supporting the functional relevance of these residues.

## RnfD subunit is only involved in electron transfer, not in sodium ion translocation

We have established that RnfG shuttles the electrons from RnfA/E to RnfD via FMN$^G$ in a process that requires conformational changes at the RnfG-A/E and RnfG/D interfaces. After the sodium ion is released to the extracellular site, our calculations suggest that the redox potential

of AE1 downshifts by *ca.* 130 mV (due to the Na$^+$ interaction with the site) that favours the electron transferred to FMN$^G$ (see Supplementary Methods, Supplementary Table 8). After the Na$^+$ release and re-oxidation of AE1, the RnfA/E dimer could transition back to an inward-facing conformation, thus closing the extracellular site and preventing RnfG from moving towards RnfA/E (Supplementary Fig. 10d).

In contrast to previous suggestions, where RnfD (NqrB) was proposed to catalyse Na$^+$ pumping[11,12,14], we find no evidence that this subunit could support sodium ion translocation. In this regard, we observed no Na$^+$ bound in the region between FMN$^D$ and RBF in the RnfD interior, in contrast to the RnfA/E dimer, which binds Na$^+$ at stable positions near the AE1 cluster at both cytosolic and extracellular sites (Fig. 3a, b, e–h, Supplementary Fig. 12). The RnfD subunit remains rather dry in the MD simulations, with water molecules binding to the cytosolic and extracellular surfaces. In stark contrast, the RnfA/E interface shows water molecules reaching the centre of the subunits (next to the AE1 cluster), from either the cytosolic or extracellular side, depending on the conformation and redox state of the cofactors (Supplementary Fig. 16). Moreover, analysis of the Na$^+$ pathways suggests that there is a continuous ion translocation pathway from the cytoplasmic to the periplasmic side across the RnfA/E dimer (Supplementary Fig. 17), whereas we are unable to identify a sodium ion pathway within RnfD subunit. We note that the proposed RnfD-based pathway involves transfer of Na$^+$ and electrons in *opposite* directions, which would require Rnf to invest twice the SMF ($2 \times \Delta\psi_{SMF}$) for each pumping stroke, as both the sodium ion and the electron move against the SMF. In contrast, we propose that Na$^+$ and electron follow the same pathway, thus thermodynamically driving the Na$^+$ translocation against the SMF by the electron transfer, which moves along the transmembrane gradient (Fig. 4a,b). Our proposed pumping mechanism thus makes the main energy transduction step less sensitive to modulations in the membrane potential (Fig. 4b, c), which could be important for establishing an effective pumping machinery under thermodynamically highly constrained conditions.

Based on recent cryo-EM and x-ray structures of the Nqr complex from *Vibrio cholerae* in different conformations[14] it was also proposed that Na$^+$ could bind to the periplasmic side of NqrB (RnfD) at two distinct positions. However, only one of the proposed Na$^+$ binding sites (I274, R275, and Y280) is conserved between Nqr and Rnf (Supplementary Table 5). In addition, two other phenylalanines were suggested to gate the Na$^+$ translocation in NqrB, but only one is conserved in RnfD (F245) (Supplementary Fig. 7b, Supplementary Table 5), and alteration of these residues indicated reduced Na$^+$-dependent voltage formation[14]. To verify these claims, we produced alanine variants of all of these residues, and tested whether the corresponding mutants did grow autotrophically. All variants supported full growth on H$_2$ and CO$_2$ with either 20 or 2 mM NaCl, indicating that the proposed residues are not essential for Na$^+$ binding or transport (Supplementary Fig. 14f, g, Supplementary Table 3). We note that the non-feasibility of a Na$^+$ translocation pathway in the closely related NqrB subunit was also suggested in previous studies[13]. Taken together, our combined analysis suggests that RnfD is not involved in the Na$^+$ translocation, but is essential for electron transfer (Figs. 1d, 2f, g). Importantly, the similarities in the membrane domain subunits (Supplementary Fig. 7) and the global molecular architecture, point to a conserved Na$^+$-pumping mechanism between Rnf and Nqr.

## Putative sodium ion translocation mechanism

Based on our combined data, we propose a putative mechanism of redox-driven Na$^+$ translocation in the Rnf complex. When Rnf operates in the forward electron transfer direction (Fd→NAD$^+$) (Fig. 4), binding of the reduced Fd to RnfB subunit stabilises the inward-facing state. The electron is transferred from Fd along the [4Fe4S] clusters of RnfB, and reaches the terminal B8 cluster in its flexible domain. Reduction of

B8 stabilises this flexible domain, which binds to the cytosolic open surface of the RnfA/E dimer in the inward-facing state. This brings the B8 and AE1 FeS clusters into close proximity, thus kinetically enabling the electron transfer to AE1. The reduction of AE1 attracts the Na$^+$ into the buried Na$^+$ binding site from the cytoplasmic side by electrostatic effects (Fig. 3e–h, Supplementary Fig. 12, Supplementary Fig. 15). Consequently, the Na$^+$ binding next to AE1 shields the charge imbalance between the electron ($e^-$) and the cation (Na$^+$) and could enable the conformational changes in the flexible TM4 of RnfA/E (Supplementary Fig. 9a–d), with the residues in the vicinity of AE1 transiently stabilising the bound Na$^+$ (Fig. 3e–g, Supplementary Fig. 12b–d). The changes in solvation and electrostatic interactions enable conformational rearrangements that result in the inward- to outward- transition of the RnfA/E dimer (Supplementary Movie 3), which in turn allows the flavin domain RnfG to bind closer to RnfA/E. The outward-facing state brings the AE1 cluster closer to the periplasmic side (Supplementary Fig. 9e, g), decreasing the AE1-FMN$^G$ distance, and enabling the electron transfer to FMN$^G$. Upon oxidation of AE1, the Na$^+$ affinity decreases, and the Na$^+$ is released to the periplasmic side. The electron on FMN$^G$ is further transferred to the flavins in RnfD, located at close *edge-to-edge* distances, and onwards to the [4Fe4S] clusters of RnfC, completing the U-shaped electron transfer pathway. Reduction of the C2 cluster next to FMN$^C$, could increase the binding affinity of NAD$^+$, which forms a stacking interaction with FMN$^C$ (*cf.* also refs. [21,23]). Binding of another reduced Fd on RnfB, transfers a second electron through the chains, following similar principles as in the first pumping stroke, and couples to the translocation of a second Na$^+$ across the membrane. The putative two-electrons reduced FMN$^C$H$^-$intermediate could then transfer the electrons to NAD$^+$, forming NADH by hydride transfer (H$^-$) reaction[21,23].

The Rnf complex thus combines key features of electrostatic coupling mechanisms that are central for redox-driven ion pumps[24,25] with an alternative access switching mechanism[26] that is observed in primary active transporters[27,28]. The conformational switching suggests that Rnf may pump Na$^+$ via a "rocker-switch"-type mechanism, with a moving boundary around the AE1 cluster, establishing alternating access around the Na$^+$ binding site that is controlled by rearrangement of the symmetrical RnfA/E subunits. Interestingly, the outward/inward transition causes the loop region between TM4a/b or TM1a/b to tilt upon movement of the AE1 cluster towards the intracellular leaflet of the membrane by around 4 Å (Fig. 3g, Supplementary Fig. 9e–g). We suggest that the gating is thus likely achieved by the combination of electrostatic and kinetic effects, where the AE1 reduction modulates the Na$^+$ affinity, which in turn controls the rate of electron transfer by changing the distance between the redox clusters, as well as their redox potentials (Figs. 2h, 4b). Despite not undergoing large conformational changes, the Na$^+$ pumping in Rnf shows interesting energetic similarities to the redox-coupled proton transfer reactions of cytochrome oxidase, where the proton transfer barriers are modulated by electric field-dependent water contacts, and the electron transfer being strictly controlled by the proton transfer reaction[29]. However, the Rnf complex employs a complete inward-outward transition (Supplementary Fig. 11, Supplementary Movie 3–5) to establish the barrier modulations, as suggested by analysis of the cryo-EM data and the global motions observed in our MD simulations and free energy calculations, thus further supporting a rocker-switch type alternative access mechanism. Our results suggest that Na$^+$ translocation is driven by the redox state of AE1, whilst further evidence to support our mechanistic model is also needed in future studies.

## Discussion

We have elucidated here key molecular principles of the redox-driven sodium ion translocation mechanism in the Rnf complex by integrating cryo-EM studies with multi-microsecond atomistic simulations and site-directed mutagenesis data. Our combined approach showed that the reduction of a membrane-bound [2Fe2S] cluster induces sodium ion binding to the RnfA/E site by an electrostatic coupling mechanism and triggers an inward-outward conformational transition around RnfA/E that pumps the sodium ions by an alternating access mechanism. Our mutagenesis experiments validated that the RnfA/E constitutes the sodium ion channel, contrary to previous suggestions, whilst our redox-controlled cryo-EM analysis and MD simulations revealed conformational changes along the exergonic and endergonic electron transfer pathways.

The proposed redox-driven alternate access mechanism introduces a unique principle for ion pumping employed by the Rnf and related enzymes that appeared early on during evolution to facilitate survival under energy-limited conditions at the thermodynamic limit of life. Nevertheless, similar molecular principles could be also employed in contemporary aerobic organisms, where the related Nqr is utilised as a respiratory machine in many important pathogens. The proposed redox-driven sodium ion pumping mechanism could thus provide a basis for drug design to combat emerging pathogens.

## Methods

### Production and purification of the Rnf complex

To purify the Rnf complex from *A. woodii*, we used a plasmid-based production system. *A. woodii* Δ*rnf* containing the pMTL84211_Ppta_ack_rnf plasmid[30] was grown in 4 L carbonate-buffered complex medium according to (ref. 31) containing 50 μg ml$^{-1}$ uracil and 5 μg ml$^{-1}$ clarithromycin on fructose as carbon and energy source at 30 °C under strictly anoxic condition. All purification steps were performed under strictly anoxic conditions at room temperature in an anoxic chamber (Coy Laboratory Products) filled with 95–98% N$_2$ and 2–5% H$_2$. Cells were harvested at stationary growth phase and washed in buffer A (50 mM Tris/HCl, pH 8, 20 mM MgSO$_4$, 2 mM dithioerythritol (DTE), 4 μM resazurin). This was followed by resuspending cells in buffer A and incubated with 3 mg ml$^{-1}$ lysozyme for 1 h at 37 °C. Afterwards, cells were disrupted using a French Pressure cell (110 Mpa) after 0.5 mM PMSF and Dnase I were added. The cell debris were removed by centrifugation (15,000 × *g* for 15 min) before cytoplasmic- and membrane fraction were separated by ultracentrifugation (210,000 × *g* for 45 min). The membrane fraction was used for the purification of the Rnf complex and resuspended in buffer A1 (50 mM Tris/HCl, pH 8, 20 mM MgSO$_4$, 5 μM FMN, 2 mM DTE, 4 μM resazurin) to a final protein concentration of 10 mg/ml. For solubilisation of membrane proteins, n-dodecyl-β-D-maltopyranoside (DDM) was added at a final concentration of 1% [w/v] and the preparation was stirred for 2 h at 4 °C. The solubilised proteins were separated from the membrane fraction by ultracentrifugation (210,000 × *g* for 45 min). The supernatant containing the solubilised proteins was diluted in buffer A2 (50 mM Tris/HCl, pH 8, 20 mM MgSO$_4$, 5 μM FMN, 0.02% DDM, 2 mM DTE, 4 μM resazurin) to decrease the initial DDM concentration to 0.5% and incubated with nickel nitrilotriacetic acid (Ni$^{2+}$-NTA) resin (Qiagen) for 1 h at 4 °C. The Rnf complex was purified according to the manufacturer´s protocol using buffer A2 for equilibration, buffer W (buffer A2 + 20 mM imidazole) for washing steps and buffer E (buffer A2 + 150 mM imidazole) for elution. Elution fractions containing Rnf were collected, pooled and further purified by gel filtration on "Superdex™ 200 Increase 10/300", which was equilibrated with buffer S (50 mM Tris/HCl, pH 8, 20 mM MgSO$_4$, 150 mM NaCl, 5 μM FMN, 0.02% DDM, 2 mM DTE, 4 μM resazurin).

### Bacterial strains and growth conditions

*Escherichia coli* HB101 (Promega, Madison, WI, USA) was used for plasmid construction and cultivated in LB medium (Bertani 1951) at 37 °C. *A. woodii* strains were grown at 30 °C in complex medium according to ref. 31. by adding 50 μg ml$^{-1}$ uracil under anoxic conditions using either 20 mM fructose or H$_2$ + CO$_2$ to an overpressure of

110 kPa as carbon and energy source. For growing under sodium depleted conditions, the medium contained only 2 mM of sodium chloride instead of 20 mM NaCl. Erythromycin for use in *E. coli* (250 μg ml⁻¹) or its derivative clarithromycin for use in *A. woodii* (5 μg ml⁻¹) was added appropriately. Growth was monitored by measuring the OD at 600 nm.

### Construction of pMTL_84211_Ppta_ack_rnf and transformation of *A. woodii*

Construction of pMTL_84211_Ppta_ack_rnf and transformation of *A. woodii* for the construction of the plasmid, genomic DNA of *A. woodii* was used as template in a PCR using the oligonucleotides 1 and 2 for the amplification of the *rnf* encoding genes, whereas pMTL84211_Ppta_ack (containing a erythromycin resistance cassette and a Gram− p15A origin of replication) was amplified with the oligonucleotides 3 and 4[30]. The amplified product was ligated via Gibson assembly to construct the plasmid pMTL84211_Ppta_ack_rnf. A His-Tag encoding sequence was added to the C-terminus of RnfG by using oligonucleotides 5 and 6 via site-directed mutagenesis (Q5® Site- Directed Mutagenesis Kit, NEB, Frankfurt am Main, Germany). The resulting plasmid was checked by sequencing analysis (Microsynth Seqlab, Göttingen, Germany) before it was transformed into the *A. woodii Δrnf* mutant[1]. Plasmids were isolated from the *A. woodii Δrnf* mutant to verify successful transformation and retransformed into *E. coli* HB101. Plasmids were again isolated and checked by sequencing analysis. The expression vector pMTL84211_Ppta_ack_rnf was used as a template to introduce targeted mutations into the open reading frame of the Rnf using the corresponding oligonucleotides (Supplementary Table 9). The introduction of targeted mutations (Supplementary Table 3) was verified by sequencing.

### Enzymatic activity measurements

The ferredoxin:NAD oxidoreductase activity was measured as described previously[32] in buffer 1 (20 mM Tris/HCl, pH 7.2, 20 mM NaCl, 2 mM DTE, 4 μM resazurin) in 1.8 ml anoxic cuvettes (Glasgerätebau Ochs, Bovenden, Germany) under a CO atmosphere at 30 °C. Before addition of the purified protein, 30 μM ferredoxin was added and pre-reduced with CO using the purified CODH/ACS from *A. woodii*[33]. The reaction was started by addition of NAD⁺.

### Analytical methods

The protein concentration was determined according to Bradford[34]. Proteins were separated in 12% polyacrylamide gels according to Schägger and von Jagow[35] and stained with Coomassie brilliant blue G250. The iron content was determined calorimetrically according to Fish et al.[36].

### Cryo-EM data acquisition, processing, and model building

For the cryo-EM structure determination of the Rnf complex, all grids were prepared using vitrobot in an anaerobic glove box with a gas composition of 95% $N_2$ and 5% $H_2$. All other operations, which included incubating Rnf with NADH or reduced ferredoxin were also performed under anoxic conditions.

To reduce the Rnf complex with NADH, the purified complex was incubated with 500 μM NADH for 10 min in an anaerobic chamber before vitrification. 3.5 mg ml⁻¹ of the incubated Rnf complex with NADH was applied to glow-discharged Quantifoil grids, blotted with force 4 for 3.5 s, and vitrified by directly plunging in liquid ethane (cooled by liquid nitrogen) using Vitrobot Mark III (Thermo Fisher) at 100% humidity and 4 °C. The data collection was performed using SerialEM[37] on a FEI Titan Krios transmission electron microscope operated at 300 keV, equipped with a Gatan K3 Summit direct electron detector. In order to avoid biasing, a slight preferred orientation, 10° tilted frames were acquired at a magnification of 60,000 x with a dose rate of 50 e⁻ per Å² which were fractionated into 50 frames at a

calibrated pixel size of 1.09 Å. A total of 20,443 images were acquired. The dataset was entirely processed in CryoSparc[38]. Initially, the frames were gain-normalised, aligned, dose-weighted using Patch Motion correction, and followed by CTF estimation. Only micrographs having CTF fits ≤3.5 Å were used for further processing. Manual picking of particles, followed by multiple rounds of topaz[39] to train a model for particle picking was used. Once the model picked all the particles from a small subset of the data, the particle extraction was performed on the entire dataset. This was verified by performing reference-free 2D classes on the extracted particles. For the NADH-reduced Rnf complex, 3.6 million particles were classified into three ab-initio classes to remove junk and non-aligning particles. Thereafter, the remaining 1.4 million particles were further cleaned by subjecting them to heterogenous refinement with forced classification. After filtering, the single class containing 912 K particles were then subjected to 3D classification. Out of the three classes generated and the two well-aligned classes with combined 645 K particles were refined using NU-refinement, resulting in a density map with a resolution of 3.3 Å. Before the final NU-refinement the particles were re-extracted with a box size twice as the particle size. This step was done in order to capture all the high-resolution signal scattered around the particles.

The structure of the Rnf complex reduced with low potential ferredoxin was determined as follows (Supplementary Fig. 3); firstly, low potential reduced ferredoxin was generated by incubating 45 μM of Fd with carbon monoxide dehydrogenase under a carbon monoxide gas environment[8]. Thereafter, this pre-reduced Fd was added to 3.5 mg ml⁻¹ (or 16 μM) of the Rnf complex, and the mixture was incubated to let the Fd bind and release electrons into the Rnf complex. All the concentrations used are according to the amounts of the protein used in the oxidoreductase activities, however, increased to make them suitable for the cryo-EM experiments. Immediately after, the reduced Rnf complex was applied to glow-discharged Quantifoil grids, blotted for 4 s with force 5, and flash-frozen into liquid ethane using Vitrobot Mark IV (Thermo Fisher) at 100% humidity and 4 °C. The anaerobically vitrified grids were then transferred to a FEI Titan Krios G4 transmission electron microscope operated at 300 keV, equipped with a Falcon 4i direct electron detector (Thermo Scientific) and an energy filter with a slit width of 10 eV. The automated data acquisition was performed using EPU software. Images were recorded with an electron dose of 60 e⁻ per Å² in counting mode at a nominal magnification of ×165,000, which resulted in a calibrated pixel size of 0.75 Å. This was followed by exporting the recorded images in an EER file format. A large dataset of 56,909 images was acquired due to the low abundance of Rnf complex on the grid. For this dataset, after multiple rounds of particle picking using topaz[39], roughly 5 million participles were subjected to five ab-initio classes to filter out the junk particles. The remaining 1.8 million particles were used for heterogeneous refinement (three classes) using forced classification and the best aligning class of 1.5 million particles was used for 3D classification without alignment (Eight classes generated but six best are shown). Out of these six classes, one of the classes had the mobile RnfB domain containing the B8 [4Fe4S] cluster was found to be closer to the membrane, whereas the other two best aligning classes had the B8 cluster away from the membrane. The class with the B8 cluster closer to the membrane contained 260,238 particles, which were refined to get a density map with a resolution of 2.9 Å. The leftover particles (one having B8 cluster domain away from membrane) were further classified to remove any poorly resolved particles. This class consisting of roughly 600 K particles, was refined with NU refinement and resulted in a density map with a resolution of 2.8 Å. To gauge the dynamics of RnfG subunit, 3D-classification was performed by placing a soft mask around the RnfG subunit. One class (in blue, see Supplementary Fig 3c) out of the total six generated classes showcased the RnfG subunit moving closer to the AE1 cluster. However, the movement was only minor and could arise from the tight binding of the detergent belt

around the membrane subunits. Further, masked local refinements on cytosolic, membrane, and periplasmic parts were performed which yielded resolutions of 2.9 Å, 2.8 Å, and 3.0 Å, respectively, which enhanced the overall quality of the individual regions. Similar to Rnf with NADH processing pipeline, before final refinements the particles were re-extracted with a large box size to cover all the delocalised signal from the particles.

For the *apo* state of the Rnf complex, the cryo-EM grids were prepared using a similar anaerobic method as described above for Rnf with reduced NADH and Fd. The protein complex at 2 mg ml$^{-1}$ was applied to glow-discharged Quantifoil grids, blotted with force 4 for 3.5 s, and vitrified by directly plunging in liquid ethane (cooled by liquid nitrogen) using Vitrobot Mark III (Thermo Fisher) at 100% humidity and 4 °C. The data was collected on a FEI Titan Krios G4 transmission electron microscope as described above with a pixel size of 0.75 Å and a dose of 60 e$^-$ per Å$^2$. Due to the cluttering of particles on the edge of the grid holes as a result of thin ice, a large dataset with a total of 39,965 images were recorded, and after filtering, only 28,000 were used for final processing. The *apo* state of the Rnf complex was determined using a similar strategy as described above for the structures of Rnf with NADH and reduced Fd. After filtering the motion corrected micrographs, particles were picked manually and then topaz model was trained to perform a final picking. Roughly 1.9 million particles were subjected to 5 ab-initio classes to remove junk particles. The ab initio class consisting of Rnf complex particles (740,701) was then used for heterogenous refinement. The well aligned 645,102 particles of Rnf were then used for the final round of NU refinement which resulted in a map attaining a resolution of 3.0 Å.

All the initial model from Rnf complex from *A. woodii* was generated separately from their protein sequences using AlphaFold[40], and thereupon fitted as rigid bodies into the density using UCSF Chimera[41] (version 1.16). The model was then manually rebuilt and refined using Coot[42] (version 0.9.8.6). The cofactors were placed manually inside the density and refined by using their generating their respective restraint CIF files. The final model was subjected to real-space refinements in PHENIX[43] (version 1.21). The CIF files for FMN (in RnfC), RBF, and NADH were generated using the electronic Ligand Builder and Optimization Workbench (eLBOW) tool within the Phenix suite, using the chemical file (mol2) and final geometry file from the PDB models as inputs. The covalent bonds of FMN$^{RnfG}$ with Thr185$^{RnfG}$, and FMN$^{RnfD}$ with Thr156$^{RnfD}$, were introduced using the AceDRG plugin in Coot software. AceDRG creates the CIF files for the covalent links generated, and additionally, we manually created parameter files to maintain the covalent link during the Phenix real space refinement. The cryo-EM map, PDB model, along with CIF files from AceDRG and parameter files were used as input to refine the structure. The models were visualised using UCSF Chimera[41] (version 1.16), UCSF ChimeraX[44] (version 1.5), and PyMOL[45] (version 1.20 open source).

## Molecular dynamics simulations

Atomistic (MD) simulations of NADH- and Fd-reduced Rnf were performed in a POPC membrane with TIP3P water, and 250 mM NaCl. The protonation states were determined using PROPKA3[46] (see Supplementary Table 7). The simulations were performed with the cofactors modelled in different redox states in an *NPT* ensemble ($T = 310$ K, $p = 1$ bar) based on the CHARMM36 force field[47] and in-house developed cofactor parameters[48]. The MD models, with *ca.* 430,000 atoms, were propagated for 16.8 μs using a 2 fs integration timestep with NAMD v. 2.14 and 3.0[49]. APBS[50], VMD[51], UCSF Chimera[41], and MDAnalysis[52] were used for analysis. See Supplementary Methods for further details about the MD simulations.

## Reporting summary

Further information on research design is available in the Nature Portfolio Reporting Summary linked to this article.

## Data availability

Cryo-EM maps are available in the Electron Microscopy Data Bank; Rnf treated with NADH (19915) [https://www.ebi.ac.uk/emdb/EMD-19915], Rnf with reduced Fd state-1 consensus map (19919) [https://www.ebi.ac.uk/emdb/EMD-19919], Rnf with reduced Fd state-2 B8 closer to membrane (19916) [https://www.ebi.ac.uk/emdb/EMD-19916], and Rnf *apo* (19920) [https://www.ebi.ac.uk/emdb/EMD-19920]. The atomic models of Rnf complex are available in the Protein Data Bank; Rnf treated with NADH (9ERI), Rnf with reduced Fd state-1 consensus map (9ERK), Rnf with reduced Fd state-2 B8 closer to membrane (9ERJ), and Rnf *apo* (9ERL). Structural and sequence data used for comparison with Rnf subunits are available in the Protein Data Bank (PDB – 7XK3, Nqr complex) [https://www.rcsb.org/structure/7XK3]. Strains and plasmids generated in this study are available from the corresponding authors upon request. MD simulations data are available in the zenodo repository [https://doi.org/10.5281/zenodo.10974499]. Source data are provided with this paper.

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

## Acknowledgements

J.M.S. acknowledges the DFG for an Emmy Noether grant (SCHU 3364/1–1) and the European Union's Horizon 2020 research and innovation programme (Two-CO2-One, grant agreement no. 101075992). Views and opinions expressed are, however, those of the author(s) only and do not necessarily reflect those of the European Union or the European Research Council. Neither the European Union nor the granting authority can be held responsible for them. This work received funding from the German Research Foundation (DFG) and the European Research Council under the European Union's Horizon 2020 research to V.M. This work was also supported by the Swedish Research Council (VR, V.R.I.K.), and the Knut and Alice Wallenberg (KAW) Foundation (grant 2019.0251, 2024.0220 V.R.I.K). Computing resources were provided by the National Academic Infrastructure for Supercomputing in Sweden (NAISS 2024/1-

28, 2023/6–128) and the Swedish National Infrastructure for Computing (SNIC 2022/1-29, 2022/13–14), and the Leibniz-Rechenzentrum (LRZ, project: pr83ro).

## Author contributions

J.R. and A.S. isolated the complex and performed the entire biochemical and functional assays along with the mutational analysis, A.K. and T.R.T. vitrified cryo-EM grids anaerobically and performed initial screening, A.K. and S.B. collected the cryo-EM data, A.K. performed the cryo-EM data analysis, model building, and refinement, A.K., J.M.S., V.R.I.K. and V.M. interpreted the cryo-EM structures, H.K., P.S. and V.R.I.K. performed MD simulations, calculated the free energies, and derived mechanistic models, A.K., H.K., P.S., J.M.S., V.R.I.K. and V.M. wrote the manuscript from the input of all other authors. A.K., J.R., H.K., P.S. contributed equally to this work.

## Funding

## Competing interests

The authors declare no competing interests.
