## [Transparent Peer Review file · Nature Communications]

Molecular principles of redox-coupled sodium pumping of the ancient Rnf machinery

Corresponding Author: Professor Volker Müller

Version 1:

Reviewer comments:

Reviewer #1

(Remarks to the Author)

Overview:

The authors presented the cryogenic electron microscopy (cryo-EM) structures of the Rnf complexes in NADH-reduced form, ferredoxin (Fd) reduced forms (two states) as well as the apo enzyme. The four resolved cryo-EM structures by themselves do not provide evidences for redox linked sodium pumping, they were then being used as the starting points for MD simulations. Remarkably, the authors observed the inward-facing and outward-facing conformations of the sodium channel, as well as the domain movement of RnfB8 and RnfG which regulates electron transfer distances to the extent that electron transfer at physiologically relevant rate becomes possible, which enabled the authors to formulate a redox-dependent sodium pumping mechanism of the Rnf complex. The authors also established in this study that, it is the RnfA/E dimer which catalyses sodium pumping, instead of RnfD as proposed in earlier studies of Rnf complexes as well as the related NqrB of the Nqr complexes.

Although the cryo-EM structure of Rnf complexes have been resolved and published recently by various groups (c.f. Zhang, L., Einsle, O. Architecture of the RNF1 complex that drives biological nitrogen fixation. *Nat Chem Biol* (2024). <https://doi.org/10.1038/s41589-024-01641-1>, Vitt, S., Prinz, S., Eisinger, M. et al. Purification and structural characterization of the Na⁺-translocating ferredoxin: NAD⁺ reductase (Rnf) complex of *Clostridium tetanomorphum*. *Nat Commun* 13, 6315 (2022). <https://doi.org/10.1038/s41467-022-34007-z>), these works focus on very detailed structural descriptions but contain rather speculative or limited mechanistic discussions. In contrast, this work provides novel energetic insights thanks to the molecular dynamics simulations and calculations of the redox potential energies of different conformational states and is of significance to understand how Rnf complex work.

Since much focus were placed on the simulations to elucidate the redox coupling mechanism, the presentation of biophysical and structural data is not satisfactory (see detailed questions). Sometimes the descriptions of the cryo-EM structures are vague or inaccurate, and the results of biophysical, biochemical experiments and cryo-EM structural analyses are often not well distinguished.

The authors provided the wwPDB preliminary validation reports. Although it is possible to justify the quality of the raw and processed structural data using the preliminary validation reports in most of cases, since several questions arose (the camera used are not consistent in the methods section and in the wwPDB preliminary validation reports for the state 1 and state 2 Fd reduced Rnf complex, large RMSZ bond and angle deviations of riboflavin were found in all four structures, the covalent linkages of FMN were missing) and the authors already obtained the PDB and EMDB codes of the four structures (entries on hold until publication), I would ask the authors to provide the wwPDB deposition validation reports with exact PDB and EMDB ID in order to eliminate the possibility of a scenario that the preliminary wwPDB validation reports contain errors which may have been corrected upon final deposition of the structural data.

Detailed Questions:

Line 96: Analysis of the cofactors revealed that the complex contained all the expected cofactors ... → Since this sentence appeared in the purification section instead of in the molecular architecture (cryo-EM) section, it is not clear to me if the

analysis refers to spectroscopic analysis of the purified protein similar to that in Nat Commun 13, 6315 (2022). (<https://doi.org/10.1038/s41467-022-34007-z>), Nat Struct Mol Biol 30, 1686–1694 (2023). (<https://doi.org/10.1038/s41594-023-01099-0>)? and if there are any data (not) shown? For instance, how was the AE1 Fe₂S₂ cluster being expected in the purified sample since references 12 and 13 are exclusively Nqr complexes (cf. line 150)?

Line 115: Since all determined structures have overall the same organisation and cofactor content, we focused our description on the Rnf complex with the bound NADH (Fig. 1). —> The authors should state in the legend of Fig. 1 that the Rnf structure shown is the NADH-bound state, because the distances between cofactors of the Fd-reduced Rnf complex are not the same as the NADH-reduced one (c.f. Fig. 2a and Line 217).

Line 119: On the cytoplasmic side, the complex binds RnfC and RnfB ... —> I do not understand the meaning of “the complex binds” as if RnfC and RnfB can dissociate?

Line 125: Extended Data Fig. 5d: —> Please label the contour level of the cryo-EM maps in r.m.s.d. of all cofactors shown as well as in other (Extended Data) Figures.

Line 127: ... unstructured N- and C-terminal loops of RnfD bind to the cavities within RnfC that stabilise the subunit. —> If the aforementioned loops are unstructured, how did the author know where they bind and which cavities they bind?

Line 134: RBF forms hydrogen-bonds through its ribityl moiety with several conserved residues (D249, N123), which could stabilise the semiquinone form of the cofactor (Fig. 2d). —> Please show the aforementioned hydrogen bonds and bond length in Fig. 2d, as well as the contour level in r.m.s.d. of RBF. After inspection the four supplied wwPDB validation reports, large RMSZ bond (around 4 Å) and angle (around 3°) deviations of riboflavin are found in all four structures. This indicates that there is an issue with the restraints used for model refinement (c.f. question for Line 576). Please explain.

Line 138: FMND is in proximity... (Fig. 2c). —>The phosphate group of FMND is not in the density, which is an indication that the FMN coordinates were not refined with covalent linkages to the attached threonine (c.f. questions for Line 879). See also: “The FMN cofactor is located at the tip of this domain, bound to residue T202G. The covalent attachment of the FMN moieties of both RnfD and RnfG requires the action of the flavinyl transferase AbpE”. Zhang, L., Einsle, O. Architecture of the RNF1 complex that drives biological nitrogen fixation. Nat Chem Biol (2024). <https://doi.org/10.1038/s41589-024-01641-1>.

Line 146: we observed at the pseudo-symmetric axis a rhombus-shaped density resembling a [2Fe₂S] cluster (Fig. 1, Fig. 2b). —> Please indicate the map contour in r.m.s.d. around AE1 in the inset of Fig. 2b.

Line 154: Although the core of RnfB shows low flexibility, its C-terminal region, as well as the B8-domain, are highly mobile, as indicated by the poor density and low local resolution (Extended Data Fig. 2f, 4f, and 5a, f, g). —> Since the Fd-like B8-domain of RnfB is highly dynamic (c.f. Line 177) and the maps in Extended Data Fig. 3 do not explicitly show the position of the B8 cluster but the B8 domain with incomplete densities, I wonder if the authors can show the ligation environment as well as the map of the B8 [4Fe₄S] cluster in both the Fd-reduced (state 1, state 2) and in the NADH-reduced Rnf complex in similar style of clusters B1-B7 in Extended Data Fig. 5f, with respective contour levels in r.m.s.d. of maps around all [4Fe₄S] centres? The justification of the modelling is essential as the B8-domain rearrangement lays the basis for the molecular dynamics simulations to explore sodium translocation.

Line 162: As shown by our functional assays (Extended Data Fig. 1c), the Rnf complex transfers electrons from the reduced Fd to NAD⁺, coupling the redox reaction to Na⁺ pumping across the membrane. —> The functional assays shown in Extended Data Fig. 1c was for the purified Rnf complex. However from the methods provided (Line 491), the activity of Rnf enzyme was assayed in a buffer without detergent, this may be suited to the assay cited in Ref. 8 wherein the Rnf complex activity was determined using the *A. woodii* membrane and I wonder how this method could work with the detergent solubilised Rnf complex? Since the purified Rnf complex did not seem to be reconstituted into proteoliposome, the authors cannot claim here that the activity shown in Extended Data Fig. 1c was coupled to Na⁺ pumping across the membrane.

Line 164: The cytosolic RnfB and RnfC subunits sit next to each other and bind to the electron carriers Fd and NAD⁺, respectively (Fig. 1c, d). —> Fd binding was not shown in Fig. 1c and 1d.

Line 217: The edge-to-edge distance between the AE1-FMNG is > 20 Å in our cryo-EM structure of the Fd-reduced state (Fig. 1d). —> Fig. 1d presents the Rnf complex with NADH bound (Line 115) thus cannot represent the aforementioned structure feature. The authors should show the arrangement and distances of all cofactors of the four cryo-EM structures in a new (supplementary) figure in the style similar to Fig. 1d so that readers can have a clear overview of the differences.

Line 246: Interestingly, our cryo-EM structure of Fd-reduced Rnf also reveals density in the well-resolved core of the RnfA/E interface that could arise from Na⁺ ions in the narrow cavity around the [2Fe₂S] cluster region (Fig. 3a), further supporting the MD data. —> Please show the densities which could arise from sodium ions with contour levels in r.m.s.d. and explain how to distinguish these densities from water molecules, c.f. Hau, J.L., Kaltwasser, S., Muras, V. et al. Conformational coupling of redox-driven Na⁺-translocation in *Vibrio cholerae* NADH:quinone oxidoreductase. Nat Struct Mol Biol 30, 1686–1694 (2023). <https://doi.org/10.1038/s41594-023-01099-0>.

Do the claimed sodium ions satisfy typical ligation coordination geometry and distances (<https://doi.org/10.1107/S0907444902003712>, and Nat Protoc 9, 156–170 (2014). <https://doi.org/10.1038/nprot.2013.172>) ? As these sodium ions are used to support the MD data, the authors should explain how did they determine that it was

sodium ions instead of water molecules which enter the RnfA/E interface in MD simulation (c.f. Line 242)?

Line 261: As discussed above, the Na⁺ affinity for the buried inward-binding site is highly sensitive to the redox state of the AE1 cluster. —> The correlation between the Na⁺ binding affinity and the redox state of the AE1 cluster was not discussed above.

Line 323: ... the RnfA/E interface shows water molecules reaching the centre of the subunits (next to the AE1 cluster), from either the cytosolic or extracellular side, depending on the conformation and redox state of the cofactors. —> How are the water molecules and the sodium ions described above (c.f. Line 246) being distinguished? Are their positions overlapping with each other?

Line 325: ... analysis of the Na⁺ pathways suggests ... —> what are the criteria to establish/ identify a sodium pathway and if any program was used?

Line 404: ... a novel principle for ion pumping employed by the Rnf and related enzymes that appeared early on during evolution. —> The authors only argued that Rnf is an ancestor of Nqr (Line 75) but evolutionary discussions are missing throughout this manuscript.

Line 406: ... fundamental principle is also employed in contemporary aerobic organisms. —> Detailed discussions and comparisons of the Nqr and Rnf redox coupling mechanisms from other organism are generally missing (in contrast, c.f. Nat Chem Biol (2024). <https://doi.org/10.1038/s41589-024-01641-1> and Nat Commun 13, 6315 (2022). (<https://doi.org/10.1038/s41467-022-34007-z>), hence the generalisation of the mechanisms proposed in the current study may not be convincing.

Line 449, 455, 459, 465, 493: —> DTE abbreviations?

Line 469: according to Heise ... —> a reference is missing?

Line 537: equipped with a Falcon 4i direct electron detector ... —> (for the Fd reduced Rnf complex) the corresponding Extended Data Fig. 3 (Line 529) shows a micrograph apparently recorded using a Gatan K3 camera with a rectangular sensor area (6k x 4k) instead of from a Falcon 4i camera with a square sensor area (4k x 4k). Furthermore, the wwPDB validation reports of the Fd reduced Rnf complex indicate that the state 1 cryo-EM micrographs were recorded using FEI Falcon 4 camera whereas the state 2 cryo-EM micrographs were recorded using a Gatan K3 camera. Since State 1 and State 2 were derived from 3D classification (Line 545), it is not reasonable that their respective raw data were recorded using different cameras.

Line 575, 577, 578: —> Please state the version number of Coot, PHENIX, UCSF Chimera, UCSF ChimeraX and PyMOL.

Line 576: The cofactors were placed manually inside the density and refined by using their generating their respective restraint CIF files. —> How the restraints CIF files were generated? using program (which?) or from a specific source?

Line 879: The FMN cofactors ... were modelled by covalently linking them to T185G and T156D via a phosphodiester bond. —> This description appears in the molecular simulation section instead of in the cryo-EM model building section. The four PDB validation reports also show no LINK records present from FMN in the coordinates. I wonder if these FMN coordinates were refined with the covalent bonds as well? If not, why?

Reviewer #2

(Remarks to the Author)

The manuscript is an in-depth study using a range of techniques to elucidate the mechanism of Rnf. I am not an expert in the Rnf family and therefore can not comment on the biology and impact in the field although it should be noted that the mechanism described looks to be a significant step forward in the field assuming the relevant literature has been cited. Overall this looks to be a robust study and credit to the authors for the work put into this. I had some minor comments on the manuscript that the authors may wish to consider;

Reference 1 does not appears until later in the manuscript.

Ln 94 should this be “allows it to” ?

Ln 96 “analysis of the co-factors...” It would be good to state what the specific analysis used here was, ie analysis by X of the co-factors to help the readers understand the robustness of the data rather than going into the methods.

For the methods a large oart relied on reference 8, it would be good for a brief overview to be given so the readers do not need to seek the methods through other sources.

line 550 rather than low resolved maybe poorly resolved would be a better phrase.

Extended data table 4, why was one data set collected at 60,000 and others at higher mag?

Figure 1 the purple and blue are close in colour and hard to distinguish especially when reproduced/smaller in print. More importantly a number of figures have red and green colours as the main colours this should be avoided where possible so those that are red/green colour blind can understand the figure.

Reviewer #3

(Remarks to the Author)

This is a beautiful work that makes a bold step toward unraveling the mechanism of the ancient respiratory enzyme Rnf of anaerobic bacterial world. The mechanism may be the oldest and a predecessor of many later respiratory enzymes such as Nqr, thus the work is highly significant. The enzyme was originally discovered as one of the key components of nitrogen fixation in *Rhodobacter* bacteria, from which it takes the name, by a surprising uphill endergonic reduction reaction, oxidizing relatively high-potential NADH. Here, the sodium motive force across the bacterial membrane helps to balance the energy. Working in reverse, the enzyme reduces NAD⁺ by a low potential ferredoxin, and pumps sodium ions across the membrane building up the ion gradient that subsequently drives ATP synthesis. The mechanism of redox-driven sodium pumping is the subject of the paper.

Although this is not the first attempt to solve the structure of Rnf by cryo-EM, and the U-shape electron transfer path was described before, the authors for the first time combine redox chemistry to produce different redox states of the enzyme and cryo-EM structural work, solving the structure for Fd-reduced and NADH-reduced enzymes, and revealing two different key structures that suggest a possible mode of action. Molecular dynamics simulations further clarify the obtained states and provide insights for Na⁺ ions binding sites, eventually suggesting (with some additional assumptions) a possible mechanism, which is clearly presented in the paper. The key mutations were explored and validated in the kinetics work. The mechanism is physically sound and surely possible, as it is similar to some of more recent redox-driven proton translocation enzymes. However, the questions remain as to what extent the data support the proposed mechanism. Some of the questions are listed below.

P2, third par, the driving force per electron $\Delta G = 450 - 320 = 130$ mV, where as it is assumed that membrane potential is 180mV; how then is the transport possible? For two Na ions the difference uphill is some 10kJ/mol – which is not a small matter, kT is 2.5, this should be clarified. Could it be that in fact ONE Na ion is pumped per 2 electrons in the chain? After all, Rnf is one ancient system, so to expect a perfectly efficient mechanism is perhaps a bit unrealistic. In fact, the opposite should be expected.

The paper could more adequately describe the previous structural work on Rnf (22), and related Rnf1. What are the key distinct new features in the structure of *A. woodii*?

The data from cryo-EM and MD are blended, which is good for model development and justification; however, it is not always clear what is presented - MD simulation results or actual data.

The two experimental structures, for Fd-reduced and NADH-reduced, should be described in greater detail. In particular, what are the differences seen around (FeS)₂ EA in the experiment, and how are they correlate with MD results?

Key question: What are the redox states produced in Fd- and NADH-reduced preparations? Fd-reduced is produced in about 1/3 concentration ratio: 16Rnf/45Fd mM whereas no data are given for NADH reduction. How many electrons are pumped in and where are they in the two states? Were Na-ions present in the preparation?

And most importantly, why is it that these two states should be intermediates in the cycle? It is clear that they may represent the initial states in the direct and reverse turnovers, but not the two different states in one turnover.

The logic in MD simulations should be explained more clearly; do we expect MD data to be more reliable than experimental structures? Why are the MD relaxed structures (most stable, I assume) not seen in the experiment?

P7, section "Redox state..." There seem to be a contradiction in the model and experiment; in the model, Fig. 4, the conformational change is induced by Na⁺ binding, no matter where the electron is, on B8 or FAD/A, whereas the key experimental finding (not fully described in the main text) that the reduction (i.e. electronic state) is responsible for conformational change. The unclear part is related to the uncertainty of the reduced states generated in the experiment.

In the mechanism, the Na⁺ binding is due to FeS/EA reduction, and in MD simulations (text) binding is induced by FeS/B8 reduction. (p6, second par from bottom)

What is the mechanism, or evidence, that Na⁺ binding is involved in conformational gating? Does MD support conformational change upon Na⁺ binding? (Or is it only Na⁺ binding depends on the preset conformation?)

In MD simulations, can you estimate how different the two states (open/closed) are in energy? How much redox energy goes into converting one state to the other? This is related to overall energetics shown in Fig. 4 (I think this is too formal calculation, as no free energy related to conformational changes are included.)

In the model, Fig. 4 energy profile, the ET driving Na⁺ translocation is uphill in energy. What is the driving force for Na

pumping (against 180 mV on the membrane) then?

The idea that the redox cycling of FeS in EA can be involved in Na⁺ translocation is clear and appealing; however, the current evidence seems to be only partial, this should be more clearly stated in the text.

Minor:

I suggest to comment on what “thermodynamic limit of life” is in Intro.

What are the charges of FeS clusters in MD simulations? The data should be included in SI. How do we know their redox potentials? Same for FAD's.

What are the conclusions from ET calculations detailed in the extended Table 8, are there correlations with experiment? Some ET are likely gated or involve PT, this would make rather drastic changes I think to the formal rates shown.

Despite many questions, it's a great work, congratulations.

Reviewer #4

(Remarks to the Author)

In their paper titled “Molecular principles of redox-coupled sodium pumping of the ancient Rnf machinery” Kumar and coworkers present high resolution cryo-EM structures of the Rnf complex from *A. woodii* in a NADH- and a Fd-reduced state. They present all-atom molecular dynamics simulations and biochemical data to explain the elusive mechanism underpinning the redox-driven Na⁺ transport. Characterization of Rnf has far-reaching implications in biochemistry, energy metabolism, and biotechnology, therefore the manuscript is both timely and relevant with potential for high impact.

Overall, the manuscript is well-organized, the conclusions are supported by the data, and the uncertain steps of the mechanism are mentioned with the appropriate care. However, the paper would benefit from discussing the existing theories in more detail, specifically in the introduction.

Here, as per my expertise, I focus on the assessment of the reported simulations. These simulations provide basis for the majority of the findings presented in the manuscript, and, when possible, are cross-checked against the cryo-EM data, and complemented by mutational studies and enzymatic activity measurements. The simulations were initiated either from the NADH- or Fd-reduced Rnf models with unspecified numbers of water molecules and POPC lipids. This is also difficult to judge from Extended Data Fig.7/b. I recommend adding these numbers to the relevant Extended Data Methods section along with a brief justification of the higher-than-physiological salt concentration. The 500 ns-long (except S15 to S20) simulations follow the standard protocol and were performed in duplicates. The trajectories were classified into outward-open and inward-open states, from which the authors showed that the NADH-reduced simulations always remain in the outward-open state while some of the Fd-reduced runs show transitions between the two states. However, there is but one combination where the simulations were identical in terms of the reduced cofactors and differed only in the initial structure, which is S5,S6 and S9,S10. To support this finding, I recommend the authors repeat the simulations with S13,S14 starting from the NADH-reduced structure. Performing these additional simulations is by no means a large computational burden and will strengthen the conclusions of the manuscript.

Finally, I detail below a list of smaller inaccuracies and typographical errors I noticed in the manuscript:

- L127: the termini of RnfC are discussed in the text but not shown in Fig.1/c.
- L203: where does the B8 domain insert into? The figures suggest the cytoplasmic funnel, but this is not written.
- L313: “(the Na⁺ interaction with the site)” should read “(due to the Na⁺ interaction with the site)”
- L404: the “proposed mechanism” and the “fundamental principle” are not stated explicitly
- L779: the eye-symbol in Fig.3/g is not used and not mentioned.
- L782: contrary to the caption, Fig.3/g is a density profile and not a radial distribution function
- L880: incorrect reference to Extended Data Fig.1/a, should be 7/a
- L1076 & L1107: unresolved abbreviation of ISC
- L1084: the mentioned RMSF data seems to be missing
- L1097: the sentence “(f) Angle of backbone... projected onto Z-axis (in Å)...” does not make sense, and the associated panel shows the angle and no distances.
- Extended Data Fig.10 : RnfB should be RnfE.

Version 2:

Reviewer comments:

Reviewer #1

(Remarks to the Author)

The authors have successfully responded all my requests and made corresponding corrections and improvements. I have no further questions. Nice work!

Reviewer #2

(Remarks to the Author)

The authors have addressed my comments and the manuscript is now more robust with the edits made from all reviewers.

Reviewer #3

(Remarks to the Author)

The authors have done an impressive work revising and improving the paper; in its present form I recommend the paper for publication.

The comments of the referees and the replies by the authors would be useful to publish as well as part of the discussion of the subject.

Reviewer #4

(Remarks to the Author)

In their response, the authors fully addressed my comments. They performed additional simulations where the initial structures differed but the reduced cofactors were the same, doubling their total simulation time. These simulations excluded the possibility that their observations were due to the reduced cofactors, strengthening their message.

Moreover, they improved the clarity of both the text and the figures. Consequently, I support the publication of the manuscript in its current form.

ANSWER TO REVIEWER COMMENTS

Reviewer #1 (Remarks to the Author):

Overview:

The authors presented the cryogenic electron microscopy (cryo-EM) structures of the Rnf complexes in NADH-reduced form, ferredoxin (Fd) reduced forms (two states) as well as the apo enzyme. The four resolved cryo-EM structures by themselves do not provide evidences for redox linked sodium pumping, they were then being used as the starting points for MD simulations. Remarkably, the authors observed the inward-facing and outward-facing conformations of the sodium channel, as well as the domain movement of RnfB8 and RnfG which regulates electron transfer distances to the extent that electron transfer at physiologically relevant rate becomes possible, which enabled the authors to formulate a redox-dependent sodium pumping mechanism of the Rnf complex. The authors also established in this study that, it is the RnfA/E dimer which catalyses sodium pumping, instead of RnfD as proposed in earlier studies of Rnf complexes as well as the related NqrB of the Nqr complexes.

Although the cryo-EM structure of Rnf complexes have been resolved and published recently by various groups (c.f. Zhang, L., Einsle, O. Architecture of the RNF1 complex that drives biological nitrogen fixation. *Nat Chem Biol* (2024). <https://doi.org/10.1038/s41589-024-01641-1>, Vitt, S., Prinz, S., Eisinger, M. et al. Purification and structural characterization of the Na⁺-translocating ferredoxin: NAD⁺ reductase (Rnf) complex of *Clostridium tetanomorphum*. *Nat Commun* 13, 6315 (2022). <https://doi.org/10.1038/s41467-022-34007-z>), these works focus on very detailed structural descriptions but contain rather speculative or limited mechanistic discussions. In contrast, this work provides novel energetic insights thanks to the molecular dynamics simulations and calculations of the redox potential energies of different conformational states and is of significance to understand how Rnf complex work.

Answer: We thank the reviewer for the positive statement and for appreciating the value of our integrative study combining experiments with simulations. Indeed, such integrative approach is crucial in elucidating mechanistic aspects of complex bioenergetic systems, with the MD simulations providing key insight into the energetics of the process. We have addressed all the questions in detail in the point-by-point answers below.

Comment:

Since much focus were placed on the simulations to elucidate the redox coupling mechanism, the presentation of biophysical and structural data is not satisfactory (see detailed questions). Sometimes the descriptions of the cryo-EM structures are vague or inaccurate, and the results of biophysical, biochemical experiments and cryo-EM structural analyses are often not well distinguished.

The authors provided the wwPDB preliminary validation reports. Although it is possible to justify the quality of the raw and processed structural data using the preliminary validation reports in most of cases, since several questions arose (the camera used are not consistent in the methods section and in the wwPDB preliminary validation reports for the state 1 and state 2 Fd reduced Rnf complex, large RMSZ bond and angle deviations of riboflavin were found in all four structures, the covalent linkages of FMN were missing) and the authors already obtained the PDB and EMDB codes of the four structures (entries on hold until publication), I would ask the authors to provide the wwPDB deposition validation reports with exact PDB and EMDB ID in order to eliminate the possibility of a scenario that the preliminary wwPDB validation reports contain errors which may have been corrected upon final deposition of the structural data.

Answer: We are thankful to the reviewer for the constructive comments. In response, we have rectified mistakes in the refinements of the PDB, included the covalent bonds between FMN and Thr residues in RnfG and RnfD subunits, improved the refinement of the riboflavin, and resubmitted all the models. In addition, we have now provided the structure validation reports with the exact PDB and EMDB IDs. The specific questions are addressed below.

Detailed Questions:

Question: Line 96: Analysis of the cofactors revealed that the complex contained all the expected cofactors ... → Since this sentence appeared in the purification section instead of in the molecular architecture (cryo-EM) section, it is not clear to me if the analysis refers to spectroscopic analysis of the purified protein similar to that in Nat Commun 13, 6315 (2022). (<https://doi.org/10.1038/s41467-022-34007-z>), Nat Struct Mol Biol 30, 1686–1694 (2023). (<https://doi.org/10.1038/s41594-023-01099-0>)? and if there are any data (not) shown? For instance, how was the AE1 Fe₂S₂ cluster being expected in the purified sample since references 12 and 13 are exclusively Nqr complexes (cf. line 150)?

Answer:

In our high-resolution cryo-EM structures, we could clearly observe the presence of a rhombus-shaped density at the pseudo-symmetric centre of RnfAE subunit. This resembles the characteristic shape of a [2Fe₂S] cluster, and differs from other FeS cofactors geometries (e.g. a cubic shape in the case of a [4Fe₄S] cluster, see Fig. 2 and Supplementary Fig. 5). Moreover, colorimetric analysis revealed the presence of 42 mol of Fe in the protein sample. Out of 42 mol of Fe, 40 Fe atoms could be assigned to the 10 [4Fe₄S] clusters in the Rnf complex, whereas the remaining 2 mol of Fe is consistent with the remaining rhombus-shaped density in the membrane integral RnfA/E subunits. The identity of the membrane integral [2Fe₂S] cluster in RnfA/E has been also suggested by a recent paper (Zhang & Einsle, 2024), which reports the structure of an Rnf complex from a nitrogen-fixating bacteria, further supporting our findings

The FMN-binding sites were identified in the sequence of the subunits RnfC, RnfD, and RnfG. RnfD and RnfG form covalent bonds with FMN. The presence FMN and RBF in the Rnf complex from *A. woodii* is supported by UV-vis spectroscopy and thin-layer chromatography reported in our previous publications:

- Kuhns, M., Schuchmann, V., Schmidt, S., Friedrich, T., Wiechmann, A., & Müller, V. The Rnf complex from the acetogenic bacterium *Acetobacterium woodii*: Purification and characterization of RnfC and RnfB. *Biochim Biophys Acta Bioenerg.* **1861**, 148263 (2020).

- Wiechmann, A., Trifunović, D., Klein, S. & Müller, V. Homologous production, one-step purification, and proof of Na⁺ transport by the Rnf complex from *Acetobacterium woodii*, a model for acetogenic conversion of C1 substrates to biofuels. *Biotechnol Biofuels* **13**, 208 (2020).

We have revised the main text to clarify the analysis of cofactors:

“The Rnf complex was produced using a plasmid-based production system in A. woodii that allows it to overexpress and purify an affinity-tagged version of the complex from its native host, as described previously (Supplementary Fig. 1)⁹. The presence of flavin cofactors in the complex has been reported previously^{9,20}, whereas we could identify flavin mononucleotide (FMN) in RnfC, RnfD, and RnfG and a riboflavin (RBF) in RnfD based on the structural data (see below). Moreover, the iron (Fe) content determined using calorimetric methods revealed that the purified complex contained 41.8 ± 1.5 mol iron/mol Rnf (Supplementary Table 2).”

Question: Line 115: Since all determined structures have overall the same organisation and cofactor content, we focused our description on the Rnf complex with the bound NADH (Fig. 1). → The authors should state in the legend of Fig. 1 that the Rnf structure shown is the NADH-bound state, because the distances between cofactors of the Fd-reduced Rnf complex are not the same as the NADH-reduced one (c.f. Fig. 2a and Line 217).

Answer: We thank the reviewer for pointing this out. We have now revised the figure and clarified in the caption that it shows only the structure of the Fd-reduced Rnf complex. In addition, we have prepared a new Supplementary Fig. 6 to showcase all resolved structural states of the Rnf complex (Fd-reduced, NADH-reduced, and the apo state) and comparing the distances of their cofactors.

“Fig. 1 | Molecular architecture of the Rnf complex. (a) Three-dimensional segmented cryo-EM density map of the Fd-reduced Rnf complex coloured by subunits. (b) Overall structure of the Rnf complex shown in cartoon representation. (c) Graphical representation of the Rnf complex showing the topology of the membrane-bound area along with the cofactors harboured by each subunit. (d) Cofactor content in the Fd-reduced Rnf complex, with edge-to-edge distances between the FeS clusters and cofactors shown in ångströms. See Supplementary Fig. 6 for a comparison of all resolved cryo-EM structures.”

Supplementary Fig. 6 | Comparison of cofactor distances in the different states of the Rnf complex resolved by cryo-EM. (a) Rnf complex reduced with ferredoxin. (b) Rnf complex reduced with NADH. (c) Rnf complex in its apo-state. All structures are represented as transparent surfaces with edge-to-edge distances between the cofactors reported in ångströms.

Question: Line 119: On the cytoplasmic side, the complex binds RnfC and RnfB ... → I do not understand the meaning of “the complex binds” as if RnfC and RnfB can dissociate?

Answer: We have revised the sentence accordingly.

“On the cytoplasmic side, the complex *comprises* RnfC and RnfB that are responsible for the interactions with the soluble electron carriers, whilst the periplasmic side harbours the flavoprotein RnfG.”

Question: Line 125: Extended Data Fig. 5d: → Please label the contour level of the cryo-EM maps in r.m.s.d. of all cofactors shown as well as in other (Extended Data) Figures.

Answer: This has been added.

Supplementary Fig. 5 | Cryo-EM density and model quality. [...] (c, d, e, f) Representative regions of the Rnf subunits and their surrounding electron density maps are shown, indicating the corresponding contour level (σ) values [...].

Question: Line 127: ... unstructured N- and C-terminal loops of RnfD bind to the cavities within RnfC that stabilise the subunit. —> If the aforementioned loops are unstructured, how did the author know where they bind and which cavities they bind?

Answer: We apologise for the confusion. The N- and C-terminal loops of RnfD are indeed unstructured as they do not show secondary structure characteristics. However, the parts of the loops that interact with RnfC can be modelled accurately due to the well-defined density, while the rest are poorly resolved in the cryo-EM density. This can be clearly seen in Supplementary Fig. 5c, where the loops are only partially resolved. We have now added a sentence to clarify this point:

“In addition, unstructured N- and C-terminal loops of RnfD bind to the cavities within RnfC that stabilise the subunit (Supplementary Fig. 5c) and could be partially resolved in our cryo-EM data.”

Supplementary Fig. 5 | Cryo-EM density and model quality. [...]

Question: Line 134: RBF forms hydrogen-bonds through its ribityl moiety with several conserved residues (D249, N123), which could stabilise the semiquinone form of the cofactor (Fig. 2d). → Please show the aforementioned hydrogen bonds and bond length in Fig. 2d, as well as the contour level in r.m.s.d. of RBF. After inspection the four supplied wwPDB validation reports, large RMSZ bond (around 4 Å) and angle (around 3°) deviations of riboflavin are found in all four structures. This indicates that there is an issue with the restraints used for model refinement (c.f. question for Line 576). Please explain.

Answer: We have now added the hydrogen bond lengths between RBF and N123/D249 residues. In addition, we have also included the map contour level in RMSD in Fig 2D.

Fig. 2 | Structural and functional characterisation of the electron transfer pathway in the Rnf complex. [...] (b, c, d) Segmented cryo-EM density encasing AE1, FMN^{RnfG}, and RBF, highlighting the residues coordinating the cofactors. The cysteine residues coordinating the AE1 cluster in (b) are shown, with the map contour RMSD level indicated. In (d), the hydrogen-bond distances between D249/N123 and RBF are indicated in ångströms, including the map contour level.

To refine our structures, we generated all the CIF files for the cofactors via the *electronic* Ligand Builder and Optimisation Workbench_(eLBOW) package within the Phenix refinement program, using the chemical (mol2) files and final coordinates of the FMN (in RnfC), RBF, and NADH. These CIF files were then used as input for performing the real-space refinement. A detailed description of the process has been added to the *Methods* section (see *Cryo-EM data acquisition, processing, and model building*):

“[...] The CIF files for FMN (in RnfC), RBF, and NADH were generated using the electronic Ligand Builder and Optimization Workbench (eLBOW) tool within the Phenix suite, using the chemical file (mol2) and final geometry file from the PDB models as inputs.”

Question: Line 138: FMND is in proximity... (Fig. 2c). —>The phosphate group of FMND is not in the density, which is an indication that the FMN coordinates were not refined with covalent linkages to the attached threonine (c.f. questions for Line 879). See also: “The FMN cofactor is located at the tip of this domain, bound to residue T202G. The covalent attachment of the FMN moieties of both RnfD and RnfG requires the action of the flavinyl transferase AbpE”. Zhang, L., Einsle, O. Architecture of the RNF1 complex that drives biological nitrogen fixation. Nat Chem Biol (2024). <https://doi.org/10.1038/s41589-024-01641-1>.

Answer: We thank the reviewer for pointing this out. We now used the AceDRG software to create a covalent bond between FMN and Thr185 in RnfG, and between FMN and Thr156 in RnfD, followed by manual refinement in COOT. AceDRG also generates the corresponding CIF files for the covalent bonds, which were directly used in the Phenix real space refinement program. Additionally, we manually created a parameter file for Phenix to ensure that the covalent links are maintained during refinement.

Revisions in the Methods section:

“[...] AceDRG creates the CIF files for the covalent links generated, and additionally, we manually created parameter files to maintain the covalent link during the Phenix real space refinement. The cryo-EM map, PDB model, along with CIF files from AceDRG and parameter files were used as input to refine the structure.”

In this context, we have also revised Fig. 2c, showing the FMN in RnfG subunit covalently bound to Thr185 residue and encased around by the cryo-EM map.

Fig. 2 | Structural and functional characterisation of the electron transfer pathway in the Rnf complex. [...] (b, c, d) Segmented cryo-EM density encasing AE1, FMN^G, and RBF, highlighting the residues coordinating the cofactors. *The cysteine residues coordinating the AE1 cluster in (b) are shown, with the map contour RMSD level indicated. In (d), the hydrogen-bond distances between D249/N123 and RBF are indicated in ångströms, including the map contour level.*

Question: Line 146: we observed at the pseudo-symmetric axis a rhombus-shaped density resembling a [2Fe2S] cluster (Fig. 1, Fig. 2b). —> Please indicate the map contour in r.m.s.d. around AE1 in the inset of Fig. 2b.

Answer: We have now added the contour level in the Fig 2b.

Fig. 2 | Structural and functional characterisation of the electron transfer pathway in the Rnf complex. [...] (b, c, d) Segmented cryo-EM density encasing AE1, FMN^G, and RBF, highlighting the residues coordinating the cofactors. The cysteine residues coordinating the AE1 cluster in (b) are shown, with the map contour RMSD level indicated. In (d), the hydrogen-bond distances between D249/N123 and RBF are indicated in ångströms, including the map contour level.

Question: Line 154: Although the core of RnfB shows low flexibility, its C-terminal region, as well as the B8-domain, are highly mobile, as indicated by the poor density and low local resolution (Extended Data Fig. 2f, 4f, and 5a, f, g). → Since the Fd-like B8-domain of RnfB is highly dynamic (c.f. Line 177) and the maps in Extended Data Fig. 3 do not explicitly show the position of the B8 cluster but the B8 domain with incomplete densities, I wonder if the authors can show the ligation environment as well as the map of the B8 [4Fe4S] cluster in both the Fd-reduced (state 1, state 2) and in the NADH-reduced Rnf complex in similar style of clusters B1-B7 in Extended Data Fig. 5f, with respective contour levels in r.m.s.d. of maps around all [4Fe4S] centres? The justification of the modelling is essential as the B8-domain rearrangement lays the basis for the molecular dynamics simulations to explore sodium translocation.

Answer: We now show in Supplementary Fig. 5f the B8 [4Fe4S]-cluster with its ligand sphere encased around the cryo-EM density (indicating contour level) for the Fd- and NADH-reduced states.

(c, d, e, f) Representative regions of the Rnf subunits and their surrounding electron density maps are shown, indicating the corresponding contour level (σ) values. (c) The helices TM1 and 4 for RnfA/E subunit were found slightly disordered with moderate density fits, indicating their flexible nature. (d) The FMN-NADH bound state was only obtained for Rnf complex with bound NADH, whereas for the Rnf in the Fd reduced state and the apo state contain only FMN. (f) The position of the B8 cluster encased around the cryo-EM map is shown for the Fd-reduced (state 1 and 2) and NADH-reduced state of the Rnf complex.

Due to the flexible nature of the RnfB domain containing the B8 cluster, it was difficult to accurately model the cluster. Therefore, we used an AlphaFold model of RnfB as a base model to manually refine

and build in all the clusters from B1-B8 using COOT. The density for the B8 cluster in the NADH-reduced state and the Fd-reduced state 1 was clearer, allowing us to model the B8 cluster effectively. In contrast, for the Fd-reduced state 2 structure where the B8 density was less clear, we used rigid body fitting followed by manual refinement in COOT to build the B8 cluster close to the membrane.

Question: Line 162: As shown by our functional assays (Extended Data Fig. 1c), the Rnf complex transfers electrons from the reduced Fd to NAD⁺, coupling the redox reaction to Na⁺ pumping across the membrane. —> The functional assays shown in Extended Data Fig. 1c was for the purified Rnf complex. However from the methods provided (Line 491), the activity of Rnf enzyme was assayed in a buffer without detergent, this may be suited to the assay cited in Ref. 8 wherein the Rnf complex activity was determined using the *A. woodii* membrane and I wonder how this method could work with the detergent solubilised Rnf complex? Since the purified Rnf complex did not seem to be reconstituted into proteoliposome, the authors cannot claim here that the activity shown in Extended Data Fig. 1c was coupled to Na⁺ pumping across the membrane.

Answer: We thank the reviewer for pointing this out. The homologous production and purification of the Rnf complex was first described in Wiechmann *et al.* (2020). Therefore, the Rnf complex activity was determined using the purified protein. The activity was measured in a buffer that did not contain detergent, whereas the purification buffer contained the detergent. In this regard, we expect that the purified complex binds some detergent molecules to the hydrophobic RnfA/E/D subunits. Indeed, we observe a detergent belt around the protein in the cryo-EM structures, as shown in Figure 1a, which supports our assumption.

We cannot claim that the activity shown in Supplementary Fig. 1c was coupled to Na⁺ pumping across the membrane. Nevertheless, we have described before (Wiechmann *et al.* (2020)) that the purified Rnf complex incorporated into proteoliposomes could catalyse Na⁺ transport coupled to ferredoxin-dependent NAD⁺ reduction. To avoid misunderstandings, the text has been corrected:

“As shown by our functional assays (Supplementary Fig. 1c), the Rnf complex transfers electrons from the reduced Fd to NAD⁺. Additionally, previous studies have shown that the redox reaction is coupled to Na⁺ pumping across the membrane⁹.”

Question: Line 164: The cytosolic RnfB and RnfC subunits sit next to each other and bind to the electron carriers Fd and NAD⁺, respectively (Fig. 1c, d). —> Fd binding was not shown in Fig. 1c and 1d.

Answer: We could not resolve the density of bound Fd in our Fd-reduced structure, most likely due to the transient interaction of Fd with the RnfB subunit. We have now clarified the sentence:

“Na⁺ pumping across the membrane⁹. The cytosolic RnfB and RnfC subunits sit next to each other and are responsible for binding the electron carriers Fd and NAD⁺, respectively (Fig. 1c, d).”

Question: Line 217: The edge-to-edge distance between the AE1-FMNG is > 20 Å in our cryo-EM structure of the Fd-reduced state (Fig. 1d). —> Fig. 1d presents the Rnf complex with NADH bound (Line 115) thus cannot represent the aforementioned structure feature. The authors should show the arrangement and distances of all cofactors of the four cryo-EM structures in a new (supplementary) figure in the style similar to Fig. 1d so that readers can have a clear overview of the differences.

Answer: We thank the reviewer for the suggestion. We have corrected Fig. 1d and included a new Supplementary Fig. 6 to show the distances of all cofactors in the different Rnf structures obtained in this study.

Fig. 1. [...] (d) Cofactor content in the *Fd*-reduced Rnf complex, with edge-to-edge distances between the FeS clusters and cofactors shown in ångströms. See Supplementary Fig. 6 for a comparison of all resolved cryo-EM structures.

Supplementary Fig. 6 | Comparison of cofactor distances in the different states of the Rnf complex resolved by cryo-EM. (a) Rnf complex reduced with ferredoxin. (b) Rnf complex reduced with NADH. (c) Rnf complex in its apo-state. All structures are represented as transparent surfaces with edge-to-edge distances between the cofactors reported in ångströms.

Question: Line 246: Interestingly, our cryo-EM structure of Fd-reduced Rnf also reveals density in the well-resolved core of the RnfA/E interface that could arise from Na⁺ ions in the narrow cavity around the [2Fe2S] cluster region (Fig. 3a), further supporting the MD data. → Please show the densities which could arise from sodium ions with contour levels in r.m.s.d. and explain how to distinguish these densities from water molecules, c.f. Hau, JL., Kaltwasser, S., Muras, V. et al. Conformational coupling of redox-driven Na⁺-translocation in *Vibrio cholerae* NADH:quinone oxidoreductase. *Nat Struct Mol Biol* 30, 1686–1694 (2023). <https://doi.org/10.1038/s41594-023-01099-0>.

Do the claimed sodium ions satisfy typical ligation coordination geometry and distances (<https://doi.org/10.1107/S0907444902003712>, and *Nat Protoc* 9, 156–170 (2014). <https://doi.org/10.1038/nprot.2013.172>)? As these sodium ions are used to support the MD data, the authors should explain how did they determine that it was sodium ions instead of water molecules which enter the RnfA/E interface in MD simulation (c.f. Line 242)?

Answer: We thank the reviewer for the suggestion. It is indeed not straightforward to distinguish the densities from water molecules or Na⁺ ions at this resolution (2.8-3 Å), but the coordination motif comprising carbonyl groups from the backbone is a typical ligation environment found in Na⁺, suggesting that a Na⁺ ion binds at the identified densities. We have included the RMSD contour values in the main text Figure 3:

Fig. 3 | RnfA/E subunits pump the sodium ion across the membrane gated by AE1 [2Fe2S] cluster. (a) Left: RnfA/E dimer containing the AE1 cluster and bound Na⁺ ion (in grey), *in the Fd-reduced cryo-EM structure*. **Right:** Close-up of the Na⁺ binding sites in the RnfA/E subunits around the AE1 cluster, *with the map contour RMSD level indicated*.

Additionally, the Na⁺ binding sites observed in our MD simulations are resolved independently of the cryo-EM data, thus further supporting our findings. To this end, we have modelled our initial MD setups with the ions (Na⁺ and Cl⁻) placed only in the solvent bulk phase in a homogeneous distribution. During the MD simulations, we observed spontaneous Na⁺ binding to the protein to the described binding sites. We have, therefore, determined the sodium binding site by the two different approaches, which support our claims. These data are shown in the Supplementary Fig. 12e and f, which represents the superposition of Na⁺ positions from MD simulations (green densities) and cryo-EM structures (grey spheres).

Supplementary Fig. 12 | [...] (e, f) Superposition of Na⁺ binding sites from cumulative MD simulation snapshots (green densities, obtained using the VMD VolMap tool for 1 frame/ns of the MD trajectory, from simulations S17/S18) and cryo-EM

structures (grey spheres). Panel (e) shows the periplasmic side of the RnfA/E subunit, and panel (f) shows the cytoplasmic side.

Question: Line 261: As discussed above, the Na⁺ affinity for the buried inward-binding site is highly sensitive to the redox state of the AE1 cluster. —> The correlation between the Na⁺ binding affinity and the redox state of the AE1 cluster was not discussed above.

Answer: We have modified the text to clarify the statement:

“Our MD simulations reveal that Na⁺ ions enter RnfA/E from the intracellular side only in the inward-facing conformation, and from the extracellular side in the outward-facing conformation (Fig. 3g, h, Supplementary Fig. 12a, b, Supplementary Movie S2), with the “buried” binding-site (see below) accessible for the Na⁺ ions only upon the reduction of the AE1 cluster, suggesting that the Na⁺ pumping is coupled to the reduction of the cluster (Fig. 3e-h, Supplementary Figs. 12b-d, 15).”

“As mentioned above, the Na⁺ affinity for the buried inward-binding site is highly sensitive to the redox state of the AE1 cluster.”

Question: Line 323: ... the RnfA/E interface shows water molecules reaching the centre of the subunits (next to the AE1 cluster), from either the cytosolic or extracellular side, depending on the conformation and redox state of the cofactors. —> How are the water molecules and the sodium ions described above (c.f. Line 246) being distinguished? Are their positions overlapping with each other?

Answer: We apologise for the unclear representations. Fig. 3 and Supplementary Fig. 16 show the cumulative positions of Na⁺ ions and water molecules during the MD trajectory (1 structure per nanosecond), while individual structural snapshots are also shown next to the ensemble representations. There is no overlap between Na⁺ ions and water molecules, as this would lead to extreme forces and velocities of the atoms that, in turn, would crash the simulations. We observe that the Na⁺ ion is partially solvated by water molecules in the exposed binding sites, while at the buried binding site, the ion partially desolvates, which is compensated by the protein electrostatics and hydrogen-bonding interactions.

Question: Line 325: ... analysis of the Na⁺ pathways suggests ... —> what are the criteria to establish/identify a sodium pathway and if any program was used?

Answer: We thank the reviewer for pointing this out, and have clarified how the sodium pathways were analysed. Initial sodium binding was determined by the probability distribution of Na⁺ ions from the AE1 cluster or the centre of RnfD subunit (see Fig 3.). Additionally, using clustering analysis (as implemented in the VolMap tool of VMD), we clustered the Na⁺ positions from all MD simulations, grouped into *inward/outward* conformations (see Supplementary Fig. 12a), and determined the occupancy percentage during the individual simulations (Supplementary Fig. 12c,d). Here, if a Na⁺ ion was located within 3 Å of a specified sidechain or backbone, the residue was considered occupied. The selection criteria is also described in the caption of Supplementary Fig. 12.

Tunnel analysis using the software CAVER shows that a continuous tunnel of ca. 1-2 Å radius forms from the cytoplasmic to the periplasmic side in the inward conformation of the RnfA/E subunit (Supplementary Fig. 17a-d). In contrast, it was not possible to identify putative Na⁺ tunnels in the RnfD subunit.

To further support this assignment, we have performed additional analysis where we tracked all Na⁺ ions entering and leaving the protein interior, and created a visual representation of the Na⁺ pathway based on these. We observe that during the inward conformation, Na⁺ ions enter from the cytoplasmic side by interacting initially with the carboxylate group of E115^{RnfE} and backbone carbonyl of P107^{RnfA}. The buried binding site is also identified as a Na⁺ cluster (shown as a red surface in Supplementary Fig. 17e), coordinated by the backbone of V106^{RnfE} and T111^{RnfA}. Similarly, during the outward conformation, the ions enter from the periplasmic side, and interact with charged and polar residues on the periplasmic surface of RnfA/E and RnfG subunits. Also here, a Na⁺ cluster is identified (Supplementary Fig. 17f), which coincides with the buried binding site with backbone interactions from L103^{RnfE}, V106^{RnfE}, and T111^{RnfA} coordinating the ion (see Supplementary Fig. 12).

Supplementary Fig. 17 | [...] (e,f) Analysis of Na^+ ion trajectories during MD simulations S15 (e) and S16 (f). The pathways followed by the ion are shown as yellow and green lines, while the Na^+ cluster detected by AQUA-DUCT¹⁶, is shown as a red surface.

Question: Line 404: ... a novel principle for ion pumping employed by the Rnf and related enzymes that appeared early on during evolution. → The authors only argued that Rnf is an ancestor of Nqr (Line 75) but evolutionary discussions are missing throughout this manuscript.

Answer: We expanded on the evolutionary link between Rnf and Nqr.

“To this end, the membrane-embedded subunits between Rnf and Nqr (RnfA/E/D/G and NqrE/D/B/C) are conserved, and suggest that the complexes may employ a similar Na^+ translocation mechanism. To operate with NADH/quinone instead of Fd/ NAD^+ as electron donor/acceptors, the Nqr complex incorporated the NqrF subunit, related to NAD(P)H ferredoxin reductase (FNR)^{16,17}, and oxidises NADH by FAD. Moreover, the soluble NqrA subunit (the equivalent to RnfC) lost its redox cofactors, while the quinone binding takes place in NqrB (equivalent to RnfD). Notably, ...

Question: Line 406: ... fundamental principle is also employed in contemporary aerobic organisms. —> Detailed discussions and comparisons of the Nqr and Rnf redox coupling mechanisms from other organism are generally missing (in contrast, c.f. Nat Chem Biol (2024). <https://doi.org/10.1038/s41589-024-01641-1> and Nat Commun 13, 6315 (2022). (<https://doi.org/10.1038/s41467-022-34007-z>), hence the generalisation of the mechanisms proposed in the current study may not be convincing.

Answer: We thank the reviewer for raising this point. The membrane domain subunits of Nqr and Rnf complexes show up to 48% of sequence identity based on our MSA (RnfA/NqrE: 37-48%, RnfE/NqrD: 29-40%, RnfD/NqrB: 22-31%), and RMSD 1.18 Å of structural similarity (see Supplementary Figure 7). The high local structural and sequence similarities suggest that the Na^+ pumping mechanism could be conserved. The main difference between Rnf and Nqr is in the electron input subunit, as they operate with different electron donors and acceptors. In this regard, the structure of the electron acceptor subunits is also conserved, but the cofactors were lost during the evolution of Nqr to operate with

quinone. Taken together, similarities in the membrane subunits and the global architecture suggest a conserved Na⁺ mechanism.

Supplementary Fig. 7 | Structural comparison of Rnf and Nqr and sequence conservation. (a) Superposition of the Rnf structure (PDB ID: 9ERI) with Nqr complex (PDB ID: 7XK3)²⁵. The overall complex aligns with an RMSD of 4.75 Å, however, the membrane integral subunits, shown as closeup, are highly similar and align with an RMSD value of 1.18 Å. **(b)** The corresponding multiple-sequence alignment (MSA) of the membrane integral subunit of Rnf complex (RnfA/E/D) and Nqr complex (NqrB/D/E). Sequence identity between Rnf and Nqr: RnfA/NqrE = 37.0-48.4%; RnfE/NqrD = 29.1-39.8%; RnfD/NqrB = 21.8-31.4%.

Revisions in the main text:

“Importantly, the similarities in the membrane domain subunits (Supplementary Fig. 7) and the global molecular architecture, point to a conserved Na⁺-pumping mechanism between Rnf and Nqr.”

Question: Line 449, 455, 459, 465, 493: → DTE abbreviations?

Answer: We have added the abbreviations for DTE.

“Cells were harvested at stationary growth phase and washed in buffer A (50 mM Tris/HCl, pH 8, 20 mM MgSO₄, 2 mM dithioerythritol (DTE), 4 μM resazurin).”

Question: Line 469: according to Heise ... → a reference is missing?

Answer: The reference has now been added.

“according to Ref. ³¹”

Question: Line 537: equipped with a Falcon 4i direct electron detector ... → (for the Fd reduced Rnf complex) the corresponding Extended Data Fig. 3 (Line 529) shows a micrograph apparently recorded using a Gatan K3 camera with a rectangular sensor area (6k x 4k) instead of from a Falcon 4i camera

with a square sensor area (4k x 4k). Furthermore, the wwPDB validation reports of the Fd reduced Rnf complex indicate that the state 1 cryo-EM micrographs were recorded using FEI Falcon 4 camera whereas the state 2 cryo-EM micrographs were recorded using a Gatan K3 camera. Since State 1 and State 2 were derived from 3D classification (Line 545), it is not reasonable that their respective raw data were recorded using different cameras.

Answer: We apologise for the confusion. The cryo-EM dataset for the Rnf Fd-reduced structure was indeed acquired on a Krios equipped with a Falcon 4i and both the states (state 1 and state 2) were obtained from the same dataset. The represented image in Supplementary Fig. 3b was obtained from a Falcon 4i camera but was cropped, hence making it look like an image acquired on a K3 camera with a dimension of 6k x 4k. We have now replaced it by an uncropped image, which has an original dimension of 4k x 4k.

Supplementary Fig. 3b

We have also rectified the mistake in the PDB deposition server and have indicated that both states were obtained from the same dataset by collecting the data on a Krios equipped with Falcon 4i camera.

Question: Line 575, 577, 578: —> Please state the version number of Coot, PHENIX, UCSF Chimera, UCSF ChimeraX and PyMOL.

Answer: We have added the version number of all the software used for structural analysis and modelling.

*“All the initial model from Rnf complex from *A. woodii* was generated separately from their protein sequences using AlphaFold⁴⁰, and thereupon fitted as rigid bodies into the density using UCSF Chimera⁴¹ (version 1.16). The model was then manually rebuilt and refined using Coot⁴² (version 0.9.8.6). The cofactors were placed manually inside the density and refined by using their generating their respective restraint CIF files. The final model was subjected to real-space refinements in PHENIX⁴³ (version 1.21) [...]. The models were visualised using UCSF Chimera⁴¹ (version 1.16), UCSF ChimeraX⁴⁴ (version 1.5), and PyMOL⁴⁵ (version 1.20 open source).”*

Question: Line 576: The cofactors were placed manually inside the density and refined by using their generating their respective restraint CIF files. —> How the restraints CIF files were generated? using program (which?) or from a specific source?

Answer: All the CIF files were generated using the eLBOW tool within the Phenix suite. The structures were first manually refined in COOT and then refined using the Phenix real space refinement. For creating the CIF file for FMN (in RnfC), RBF, and NADH, the chemical file (mol2) and final geometry file from the PDB models were subjected to the eLBOW tool within the Phenix suite. A detailed description in the methods part has now been added.

Revision in the Methods section:

“[...] The CIF files for FMN (in RnfC), RBF, and NADH were generated using the electronic Ligand Builder and Optimization Workbench (eLBOW) tool within the Phenix suite, using the chemical file (mol2) and final geometry file from the PDB models as input. The covalent bonds of FMN^{RnfG} with Thr185^{RnfG}, and FMN^{RnfD} with Thr156^{RnfD}, were introduced using the AceDRG plugin in Coot software. AceDRG creates a CIF files for the covalent links generated, and additionally, we manually created parameter files to maintain the covalent link

during the Phenix real space refinement. The cryo-EM map, PDB model, along with CIF files from AceDRG and parameter files were used as an input to refine the structure.”

Question: Line 879: The FMN cofactors ... were modelled by covalently linking them to T185G and T156D via a phosphodiester bond. —> This description appears in the molecular simulation section instead of in the cryo-EM model building section. The four PDB validation reports also show no LINK records present from FMN in the coordinates. I wonder if these FMN coordinates were refined with the covalent bonds as well? If not, why?

Answer: As discussed above, we created covalent bonds between FMN with T185 (in RnfG) and T156 (in RnfD) residues in all cryo-EM structures of the Rnf complex using AceDRG. The AceDRG generated a CIF file for the covalent links and, in addition, we manually created parameter files to maintain the covalent bonds during structure refinement. The CIF file and parameter file, along with the PDB model and cryo-EM map, were used as input for the real space refinement in the Phenix software package.

Revision in the Methods section:

“[...]The covalent bonds of FMN^{RnfG} with Thr185^{RnfG}, and FMN^{RnfD} with Thr186^{RnfD}, were introduced using the AceDRG plugin in Coot software. AceDRG creates a CIF files for the covalent links generated, and additionally, we manually created parameter files to maintain the covalent link during the Phenix real space refinement.”

Reviewer #2 (Remarks to the Author):

The manuscript is an in-depth study using a range of techniques to elucidate the mechanism of Rnf. I am not an expert in the Rnf family and therefore can not comment on the biology and impact in the field although it should be noted that the mechanism described looks to be a significant step forward in the field assuming the relevant literature has been cited. Overall this looks to be a robust study and credit to the authors for the work put into this. I had some minor comments on the manuscript that the authors may wish to consider;

Answer: We thank the reviewer for the positive assessment of our work and for recognizing its significance. We are grateful for the comments that helped us to further improve the manuscript. The questions have been addressed below.

Question: Reference 1 does not appear until later in the manuscript.

Answer: We thank the reviewer for pointing this out. Reference 1 appears for the first time in the abstract:

“Rnf is widespread in primordial organisms and the evolutionary predecessor of the Na⁺-pumping NADH-quinone oxidoreductase (Nqr)¹.”

We will await for advice from the Nature Communications editorial office, if they wish to renumber the references.

Question: Ln 94 should this be “allows it to” ?

Answer: Thank you, we have corrected the sentence accordingly.

“The Rnf complex was produced using a plasmid-based production system in A. woodii that allows it to overexpress and purify an affinity-tagged version of the complex from its native host, as described previously (Supplementary Fig. 1)⁹.”

Question: Ln 96 “analysis of the co-factors....” It would be good to state what the specific analysis used here was, ie analysis by X of the co-factors to help the readers understand the robustness of the data rather than going into the methods.

Answer: The content of moles of iron present in the complex was measured by colorimetric analysis as established in reference (W. Fish, 1988). The presence of the flavin cofactors in the A. woodii Rnf complex was characterised by UV-vis spectroscopy and thin-layer chromatography in previous publications:

- Kuhns, M., Schuchmann, V., Schmidt, S., Friedrich, T., Wiechmann, A., & Müller, V. The Rnf complex from the acetogenic bacterium *Acetobacterium woodii*: Purification and characterization of RnfC and RnfB. *Biochim Biophys Acta Bioenerg.* **1861**, 148263 (2020).

- Wiechmann, A., Trifunović, D., Klein, S. & Müller, V. Homologous production, one-step purification, and proof of Na⁺ transport by the Rnf complex from *Acetobacterium woodii*, a model for acetogenic conversion of C1 substrates to biofuels. *Biotechnol Biofuels* **13**, 208 (2020).

We have revised the main text to clarify the analysis:

*“The Rnf complex was produced using a plasmid-based production system in A. woodii that allows it to overexpress and purify an affinity-tagged version of the complex from its native host, as described previously (Supplementary Fig. 1)⁹. **The presence of flavin cofactors in the complex has been reported previously^{9,20}, whereas we could identify flavin mononucleotide (FMN) in RnfC, RnfD, and RnfG and a riboflavin (RBF) in RnfD based on the structural data (see below).** Moreover, the iron (Fe) content determined using calorimetric methods revealed that the purified complex contained 41.8 ± 1.5 mol iron/mol Rnf (Supplementary Table 2).”*

Question: For the methods a large part relied on reference 8, it would be good for a brief overview to be given so the readers do not need to seek the methods through other sources.

Answer: We have now revised the methods section, by including the missing details:

*“To purify the Rnf complex from *A. woodii*, we used a plasmid-based production system. *A. woodii* Δ rnf containing the pMTL84211_Ppta_ack_rnf plasmid³⁰ was grown in 4 L carbonate-buffered complex medium according to (Ref³¹) containing 50 μ g ml⁻¹ uracil and 5 μ g ml⁻¹ clarithromycin on fructose as carbon and energy source at 30°C under strictly anoxic condition. All purification steps were performed under strictly anoxic conditions at room temperature in an anoxic chamber (Coy Laboratory Products) filled with 95-98% N₂ and 2-5% H₂. Cells were harvested at stationary growth phase and washed in buffer A (50 mM Tris/HCl, pH 8, 20 mM MgSO₄, 2 mM dithioerythritol (DTE), 4 μ M resazurin).”*

Question: line 550 rather than low resolved maybe poorly resolved would be a better phrase.

Answer: We have corrected the phrase accordingly:

“The leftover particles (one having B8 cluster domain away from membrane) were further classified to remove any poorly resolved particles”

Question: Extended data table 4, why was one data set collected at 60,000 and others at higher mag?

Answer: The data for the Rnf reduced with NADH structure was acquired on a Krios 1 microscope armed with a K3 direct electron detector camera. The collection was performed at 60,000 X magnification, which corresponds to a physical pixel size of 1.09 Å. In contrast, all the other data, which includes the Rnf complex reduced with Fd and the apo state, was acquired on a Krios G4 microscope with a Falcon 4i camera. For Fd-reduced and apo states, the data acquisition took place in a super-resolution mode with a magnification of 165,000 X, which has a pixel size of 0.76 Å.

This has now been included in the Supplementary Table 4:

Supplementary Table 4 | Cryo-EM data collection, refinement, and validation statistics.

	Rnf NADH bound state (EMDB - 19915) (PDB - 9ERI)	Rnf Fd-reduced State-1 (consensus map) (EMDB - 19919) (PDB - 9ERK)	Rnf Fd-reduced State-2 (B8 closer to membrane) (EMDB - 19916) (PDB - 9ERJ)	Rnf apo state (EMDB - 19920) (PDB - 9ERL)
Data collection and processing				
Magnification	60,000 x ^a	165,000 x ^b	165,000 x ^b	165,000 x ^b

a: Data acquired on a Krios 1/ K3 direct electron detector camera, pixel size = 1.09 Å.

b: Data acquired on a Krios G4/ Falcon 4i camera, pixel size = 0.76 Å.

Question: Figure 1 the purple and blue are close in colour and hard to distinguish especially when reproduced/smaller in print. More importantly a number of figures have red and green colours as the main colours this should be avoided where possible so those that are red/green colour blind can understand the figure.

Answer: We thank the reviewer for pointing this out. We have modified Figure 1 to enhance the distinction between the blue and purple hues. We have also tried to make the green and red figures with different brightness so they can be distinguishable in black and white.

Revised Figure 1:

Reviewer #3 (Remarks to the Author):

This is a beautiful work that makes a bold step toward unraveling the mechanism of the ancient respiratory enzyme Rnf of an anaerobic bacterial world. The mechanism may be the oldest and a predecessor of many later respiratory enzymes such as Nqr, thus the work is highly significant. The enzyme was originally discovered as one of the key components of nitrogen fixation in *Rhodobacter* bacteria, from which it takes the name, by a surprising uphill endergonic reduction reaction, oxidizing relatively high-potential NADH. Here, the sodium motive force across the bacterial membrane helps to balance the energy. Working in reverse, the enzyme reduces NAD⁺ by a low potential ferredoxin, and pumps sodium ions across the membrane building up the ion gradient that subsequently drives ATP synthesis. The mechanism of redox-driven sodium pumping is the subject of the paper.

Although this is not the first attempt to solve the structure of Rnf by cryo-EM, and the U-shape electron transfer path was described before, the authors for the first time combine redox chemistry to produce different redox states of the enzyme and cryo-EM structural work, solving the structure for Fd-reduced and NADH-reduced enzymes, and revealing two different key structures that suggest a possible mode of action. Molecular dynamics simulations further clarify the obtained states and provide insights for Na⁺ ions binding sites, eventually suggesting (with some additional assumptions) a possible mechanism, which is clearly presented in the paper. The key mutations were explored and validated in the kinetics work. The mechanism is physically sound and surely possible, as it is similar to some of more recent redox-driven proton translocation enzymes. However, the questions remain as to what extent the data support the proposed mechanism. Some of the questions are listed below.

Answer: We are thankful to the Reviewer for their positive remarks and appreciating the significance of our study, and for their valuable suggestions to improve our work. We have addressed all the points below.

Question: P2, third par, the driving force per electron $\Delta G = 450 - 320 = 130$ mV, where as it is assumed that membrane potential is 180mV; how then is the transport possible? For two Na ions the difference uphill is some 10kJ/mol – which is not a small matter, kT is 2.5, this should be clarified. Could it be that in fact ONE Na ion is pumped per 1 electron in the chain? After all, Rnf is one ancient system, so to expect a perfectly efficient mechanism is perhaps a bit unrealistic. In fact, the opposite should be expected.

Answer: We apologise for the confusion. Indeed, the redox difference of -130 mV corresponds to the transfer of one electron, whereas the free energy difference ($\Delta G_0'$) of -6 kcal mol⁻¹ refers to the two-electron transfer reaction, leading to two Na⁺ ions pumped (when SMF < 130 mV), so the stoichiometry is one Na⁺ pumped per one electron. This has been clarified in the text.

The free energy ($\Delta G_0'$) of the Rnf-catalysed reaction is rather small, ca. -6 kcal mol⁻¹ (-25 kJ mol⁻¹ / -260 mV for 2 e⁻; -130 mV per electron), which can be used to pump up to two Na⁺ ions across the cytoplasmic membrane at a SMF of < 130 mV.

Question: The paper could more adequately describe the previous structural work on Rnf (22), and related Rnf1. What are the key distinct new features in the structure of *A. woodii*?

Answer: Thank you for the suggestion. We have now added a paragraph comparing the structure of the Rnf complex from *A. woodii* and other organisms. The Rnf complex from different species differ mainly on the cofactor arrangement in the Fd-binding subunit RnfB. Moreover, the previous Rnf structures have partially resolved structural features, as e.g. the flexible [4Fe4S] cluster domain in RnfB (our B8-domain in the *A. woodii* structure).

*"In comparison to previous Rnf structures, the Rnf complex from *A. woodii* contains the largest RnfB subunit, which harbours eight [4Fe4S] clusters, in contrast to other isoforms that contain three to six [4Fe4S] clusters^{12,21}. Moreover, the structure of the Rnf complex from *Clostridium tetanomorphum* was poorly resolved, and the resolution was insufficient to identify the membrane-bound [2Fe2S] cluster¹². In our structure, we could clearly resolve the [2Fe2S] cluster bound in the membrane, in*

complete agreement with other well resolved cryo-EM structures of Nqr and Rnf complexes^{13,14,21}. Additionally, the recent RnfI structure from *Azotobacter vinelandii*²¹ has only a partially resolved RnfB."

Moreover, in the Fd-reduced form, we could resolve a state where the B8 domain of the complex is closer to the membrane-bound AE1 cluster. This state has not been resolved by other studies of the Rnf complex. In this regard, the conformation differs from our NADH-reduced structure, where the B8 domain is closer to the B7 cluster.

Question: The data from cryo-EM and MD are blended, which is good for model development and justification; however, it is not always clear what is presented - MD simulation results or actual data.

Answer: We have now revised the figure legends to clarify the panels that refer to the MD or cryo-EM data, respectively.

“Fig. 2 | Structural and functional characterisation of the electron transfer pathway in the Rnf complex. (a) The B8 domain in RnfB shows different conformations in the NADH-reduced and Fd-reduced cryo-EM structures. (b, c, d) Segmented cryo-EM density encasing AE1, FMN^G, and RBF, highlighting the residues coordinating the cofactors. The cysteine residues coordinating the AE1 cluster in (b) are shown, with the map contour RMSD level indicated. In (d), the hydrogen-bond distances between D249/N123 and RBF are indicated in ångströms, including the map contour level. (e) Superimposed structures of RnfA/E from MD simulations classified in the inward (green) and outward (orange) conformations [...]. Edge-to-edge distances of cofactors B8-AE1 (top), and AE1-FMN^G (bottom) from MD simulations initiated from Fd-reduced Rnf structure. (f) Analysis from MD simulations showing the distance correlation between the conserved gating residues L108^{RnfA} and L22^{RnfE}, and the B8 and AE1 clusters. The MDs are clustered by the inward (green) or outward (orange) conformation.”

Fig. 3 | RnfA/E subunits pump the sodium ion across the membrane gated by AE1 [2Fe2S] cluster. (a) Left: RnfA/E dimer containing the AE1 cluster and bound Na⁺ ion (in grey), in the Fd-reduced cryo-EM structure. Right: Close-up of the Na⁺ binding sites in the RnfA/E subunits around the AE1 cluster, with the map contour RMSD level indicated. (b) Top: Cryo-EM structure of the R67^{RnfE}-E115^{RnfE} ion pair with a Na⁺ ion bound close to the glutamate residue with the densities shown in grey transparent surface. Bottom: Comparison of the R67^{RnfE}-E115^{RnfE} ion pair conformation between different states from both cryo-EM and MD simulation structures shows similarities. [...] (c, d) Structural snapshots from MD simulations (S15, S16), showing sodium binding in the intracellular (e) and extracellular (f) buried binding sites within RnfA/E (green/red) near the AE1 cluster. The binding is stabilised by backbones of V106^{RnfA}, L103^{RnfE}, and T111^{RnfA}, as well as by N112^{RnfA} side chain in the intracellular side. The perspective of the MD snapshots is indicated in panel g. (g) Dynamic ensemble of Na⁺ ions (lime-green spheres) in the membrane domain (RnfA, RnfE, RnfD) from two different MD simulations (Inward: S15, Outward: S16; see Supplementary Table 6). Inset: The sodium ions interact with the backbone carbonyls of the T111^{RnfA} and V106^{RnfE} in the inward (green) and outward (orange) conformations. (h) Probability density of sodium ion from the RnfA/E (left) and RnfD (right) centres during the MD simulations. The centre of the distribution is set to the AE1 cluster for RnfA/E and M246^{RnfD} for RnfD. All MD simulations clustered into inward (green) / outward (orange) conformations were included in the analysis.

“Remarkably, the dimeric RnfA/E interface relaxes into two distinct conformations during the MD simulations, with important implications for the Na⁺ translocation mechanism. [...]. During the conformational changes, the B8-AE1 distance decreased from ca. 20 Å (from the cryo-EM structure) further to 15-17 Å in our MD simulations (Fig. 2e, h, Supplementary Fig. 10a, c), bringing the two cofactors into feasible electron transfer distances in the inward conformation. These conformational changes are enabled by the insertion of the B8-domain (residues 30-100 of RnfB, see Supplementary Fig. 8e, f) between the RnfA/E subunits together with conformational changes in TM1/4 of RnfA”

“When accounting for the observed conformational changes during our MD simulations, our kinetic simulations predict that the electron is transferred across the complex with an overall rate of milliseconds (Fig. 4c, Supplementary Table 8). The proposed pumping model (Fig. 4a) also operates efficiently at 180 mV SMF (Fig. 4b, c, Supplementary Fig. 13)”

Question: The two experimental structures, for Fd-reduced and NADH-reduced, should be described in greater detail. In particular, what are the differences seen around (FeS)₂ EA in the experiment, and how are they correlate with MD results?

Answer: To explain the difference of Fd- and NADH-reduced states, we have introduced a new Supplementary Fig. 6, which shows the cofactor content and their edge-to-edge distances. As mentioned earlier, the most significant difference between the two states is the redox-state dependent

movement of the B8-cluster domain. The Fd-reduced state shows that the B8 cluster is closer to the membrane-bound AE1 cluster compared to the NADH-reduced state, which is located instead closer to the B7 cluster. The position of the B8-cluster in NADH-reduced and apo state of the Rnf complex is similar.

Supplementary Fig. 6 | Comparison of cofactor distances in the different structural states of the Rnf complex by cryo-EM. (a) Rnf complex reduced with ferredoxin. (b) Rnf complex reduced with NADH. (c) Rnf complex in the apo-state. All structures are represented as transparent surfaces with the edge-to-edge distances between the cofactors reported in ångströms.

In our cryo-EM structures, we could not see differences around the AE1 cluster in any of the three resolved states. However, as the Fd-reduced complex could be refined to a high resolution, we could observe non-proteinaceous density coordinated by conserved residues which may correspond to sodium ions or ordered water molecules.

Additionally, we observe significant differences in our MD simulations started either from the NADH-reduced or the Fd-reduced structures. While the simulations started from the NADH-reduced structure only sampled the *outward* conformation, resulting in large B8-AE1 cluster distance and Na⁺ ion binding only on the periplasmic side of the RnfA/E subunits, the simulations initiated from the Fd-reduced structure resulted in a mixture of the *inward* and *outward-facing* conformations (Supplementary Fig. 9). In the *inward-facing* conformation, the B8-domain inserts deeper into the membrane domain, decreasing the distance between B8 and AE1. This shifts the broken helix TM4a of the subunit RnfA (see Supplementary Fig. 7c,d and Supplementary Fig. 9c), making the buried Na⁺-binding site solvent accessible from the bulk (Supplementary Fig. 11).

Question: Key question: What are the redox states produced in Fd- and NADH-reduced preparations? Fd-reduced is produced in about 1/3 concentration ratio: 16Rnf/45Fd mM whereas no data are given for NADH reduction. How many electrons are pumped in and where are they in the two states? Were Na-ions present in the preparation?

Answer: We thank the reviewer for pointing this out. Unfortunately, we cannot characterise the redox states of the enzyme or any of its cofactors when the preparation is reduced with NADH or Fd. This would require EPR experiments, which are outside the scope of our present work.

The Fd-reduced state is prepared in a 1/3 concentration ratio to increase the chances of Fd binding and reducing the complex. As the Fd binds and transfers the electron to the Rnf, we may capture the Fd-bound Rnf complex and/or the Rnf structure undergoing distinct conformation changes as compared to the NADH-reduced and apo states. The current understanding is that a single Fd carries and transfers one electron to the Rnf complex. However, as discussed above, we cannot determine the redox state of the cofactors in the Fd-reduced state of the Rnf complex. Nevertheless, as the electron transfer from Fd is exergonic, the electrons could transfer to the FMN in RnfC subunit by inducing the conformational changes in B8 domain and RnfG subunit, which we also report. The cryo-EM data also aligns well with the biochemical assays, where we show that the Rnf complex catalyses the Fd:NAD⁺ oxidoreductase activity (electron transfer from Fd to NAD⁺) without a chemiosmotic gradient. In contrast, the NADH:Fd oxidoreductase activity cannot be measured, as this would require an external Na⁺ gradient to power the reaction.

For the NADH-reduced state, we added 500 μM NADH to the Rnf complex and incubated the sample for 10 minutes before vitrification. We have included this in the Methods section, *Cryo-EM data acquisition, processing, and model building*:

“To reduce the Rnf complex with NADH, the purified complex was incubated with 500 μM NADH for 10 minutes in an anaerobic chamber before vitrification.”

Excess amount of NADH was used in order to ensure that NADH is bound to the Rnf. The NADH oxidation is expected to transfer two electrons within the RnfC subunit. Since we observe a full occupancy of NADH/NAD⁺ cofactor in the structure, with no indication of the cofactor binding/unbinding, we, therefore, expect that only two electrons are present in the system. We cannot determine the location of these electrons, but we do not expect that they are transferred further from the FMN in RnfG to the AE1 cluster, as they are separated by > 20 Å. The electron transfer reaction from NADH to Fd is endergonic, and cannot transduce energy to trigger conformational changes as seen in the Fd-reduced state.

The buffer composition of the Rnf complex in all the cryo-EM experiments contained 150 mM of NaCl.

“Elution fractions containing Rnf were collected, pooled and further purified by gel filtration on “Superdex™ 200 Increase 10/300”, which was equilibrated with buffer S ((50 mM Tris/HCl, pH 8, 20 mM MgSO₄, 150 mM NaCl, 5 μM FMN, 0.02% DDM, 2 mM DTE, 4 μM resazurin).”

Question: And most importantly, why is it that these two states should be intermediates in the cycle? It is clear that they may represent the initial states in the direct and reverse turnovers, but not the two different states in one turnover.

Answer: We thank the reviewer for pointing this out. We have clarified that our structural snapshots represent states during the direct and inverse electron transfer conditions.

*“We obtained two structural snapshots of the Rnf complex **that represent states along the reverse and direct electron transfer directions** during catalytic turnover conditions: (1) an NADH-reduced state resolved at 3.3 Å resolution, and (2) a Fd-reduced state at 2.8 Å resolution, with the Fd purified from *Clostridium pasteurianum* and reduced by catalytic amounts of a CO dehydrogenase obtained from *A. woodii* (Supplementary Fig. 2-6, Supplementary Table 4).”*

*“Remarkably, in the Fd-reduced state of the complex, the Fd-like domain becomes highly dynamic relative to the NADH-treated sample, with sub-classification allowing us to identify distinct intermediates (Supplementary Fig. 3). This alternate conformation brings the B8 cluster to a closer proximity (20 Å) of the AE1 cluster (Fig. 2a), which is located within the membrane region of the protein (see below). **The structure of the Rnf complex with the B8 cluster closer to the membrane shows a possible intermediate state, induced by the reduction from Fd that has not been determined in previous studies.** In addition, 3D masked classification on RnfG revealed a state where the subunit and its cofactor FMN^G showed a minor movement, slightly reducing the distance to the AE1 cluster (Supplementary Fig. 3c), but also suggesting that the conformational space is hindered by the tight binding of the detergent belt (see methods).”*

Question: The logic in MD simulations should be explained more clearly; do we expect MD data to be more reliable than experimental structures? Why are the MD relaxed structures (most stable, I assume) not seen in the experiment?

Answer: It is difficult to address the exact reasons why the MD simulations and the cryo-EM structures capture different states. One key difference is that the MD simulations probe the behaviour of Rnf in a biological lipid membrane, while the cryo-EM characterisation is performed in a detergent-solubilised state. This can affect the energetics of the *inward/outward* states. However, the techniques are also highly complementary and allow us to explore different aspects of the catalytic cycle, where each method alone may provide only partial results.

Our MD simulations allow us to probe well-defined intermediates along the catalytic cycle that cannot be observed in the experiments, due to the transient nature of the states. One such example is the AE1 reduced or oxidised state. In this regard, the MD simulations initiated from the Fd-reduced structure resulted in sampling of both *outward* and *inward*-facing conformations, whereas the *outward*-facing state was not observed in the cryo-EM structures. We also note that cryo-EM captures equilibrium state populations at timescales that are inaccessible to the current MD simulation approaches. Thus, by starting from different cryo-EM structures, our MD simulations relax into a different set of conformations that mimic the transient state during turnover. It is likely that these microstates, observed during the MD simulations, average out in the different reduced states (Fd vs NADH), and leading to the refined high-resolution structures.

We have briefly commented on some differences:

"experimentally determined cryo-EM structures as starting points (see Methods, see Supplementary Table 6, Supplementary Fig. 8a-f), but embedded in a model of a biological membrane system. The techniques are highly complementary, but the MD simulations allow us to further probe the dynamics of transient redox states during the catalytic cycle that are challenging to experimentally isolate."

Question: P7, section "Redox state..." There seem to be a contradiction in the model and experiment; in the model, Fig. 4, the conformational change is induced by Na⁺ binding, no matter where the electron is, on B8 or FAD/A, whereas the key experimental finding (not fully described in the main text) that the reduction (i.e. electronic state) is responsible for conformational change. The unclear part is related to the uncertainty of the reduced states generated in the experiment.

Answer: We apologise for the unclear formulation. The conformational change observed in the B8 domain of the Fd-reduced state is due to the reduction of the B8 [4Fe4S] cluster, where the electron is transferred from Fd. We term this state as the redox-driven conformational change. In turn, the reduction of the AE1 [2Fe2S] cluster electrostatically attracts the sodium ion, coupling the electron transfer reaction with the *inward/outward* transition or Na⁺ transport. We suggest that the binding of sodium to the reduced AE1 cluster now triggers an *inward-outward* movement of the helices in RnfA/E. We have now clarified this in the legend of Figure 4:

"Fig. 4 | Redox-driven Na⁺ translocation mechanism in the Rnf complex. [...] iv: After Na⁺ binding next to AE1, RnfA/E undergoes an inward/outward transition that, in turn, moves RnfG closer to RnfA/E. v: Electron transfer from AE1 to FMN^G lowers the binding affinity for Na⁺, and results in the Na⁺ release to the extracellular side."

Question: In the mechanism, the Na⁺ binding is due to FeS/EA reduction, and in MD simulations (text) binding is induced by FeS/B8 reduction. (p6, second par from bottom)

Answer: We have clarified the text accordingly:

"Importantly, reduction of the AE1 cluster by B8 in the inward-open state leads to sodium binding near the AE1 centre (Fig. 3e-h, Supplementary Fig. 12). A conserved lysine residue (K32) together with other nearby charged residues (R154, E158) of RnfA, interact with the B8 cluster and could aid in sensing its redox changes (Supplementary Fig. 10a, b, and see below)."

Question: What is the mechanism, or evidence, that Na⁺ binding is involved in conformational gating? Does MD support conformational change upon Na⁺ binding? (Or is it only Na⁺ binding depends on the preset conformation?)

Answer: Our combined cryo-EM and MD data suggest that the Na⁺ binding depends on the conformation of the RnfA/E subunit as well as the redox state of the AE1 cluster (see below). In this regard, our simulations suggest that the cytoplasmic binding site is accessible when the protein is in an *inward* conformation, whereas the periplasmic site becomes accessible in the *outward* conformation. We propose that the conformational state is driven by the redox state of the bound cofactors. Specifically, the electron transfer from B1 to B8, with the reduction of the B8 cluster drives the transition to the *inward* conformation, as suggested by the sampling of this conformation only in MD simulations started from the Fd-reduced cryo-EM structure.

Based on our free energy calculations (see below), it is likely that the carbonyl groups of the backbone residues coordinating the Na⁺ ion in the buried site undergo a conformational transition that support the Na⁺ movement from the *inward*- to the *outward*-state. This complex process couples to the overall motion of the TM4 helices observed during the transition.

Question: In MD simulations, can you estimate how different the two states (open/closed) are in energy? How much redox energy goes into converting one state to the other? This is related to overall energetics shown in Fig. 4 (I think this is too formal calculation, as no free energy related to conformational changes are included.)

Answer: We thank the reviewer for the suggestions. The transition between the states is a highly complex process involving multiple degrees of freedom that result in the coupled motion of the RnfA/E helices, cofactors (e.g. B8-AE1 and AE1-FMN^G), and binding of Na⁺ ion movement. Exploration of this process requires extensive and computationally demanding advanced sampling methods.

To probe the energetics of the switching process, we optimised a 2D free energy landscape of the Na⁺ transport during the *inward/outward* transition in different redox states of the AE1 cluster using the string method (Supplementary Fig. 19), based on steered molecular dynamics that were used to extract initial structures along the pathways (Supplementary Fig. 18). The free energy calculations provide a putative Na⁺ transport pathway across the RnfA/E subunits, an estimation of the free energy of the *inward-to-outward* transition, and how this couples to the redox states of the AE1 cluster.

The simulations show that the *inward-to-outward* transition is energetically significantly more favoured when the AE1 cluster is reduced, suggesting that the electron transfer and sodium motion to the putative binding site is coupled to the conformational transition. In contrast, when the AE1 centre is oxidised, the *outward* conformation is endergonic and the transport barrier is significantly higher. The free energy simulations identify a continuous sodium pathway from the intracellular to the extracellular side, with free energy profiles shown in Supplementary Figure 19.

The results and analysis of the new calculations are shown in the main text and supplementary figures:

Addition in the main text:

"To estimate the energetics of the inward-outward transition, we performed free energy calculations using a variant of the string simulation method (see Supplementary Methods). Our free energy calculations suggest that during the conformational transition, a sodium ion follows a continuous pathway from the cytosolic to the extracellular site, in a process where the ion coordinates to the carbonyl group of the T111^{RnfA}, V106^{RnfE}, and L103^{RnfE} backbone, next to the AE1 cluster (Fig. 4d,e, Supplementary Fig. 20). The motion of the TM4a helix couples to the rotation of these carbonyl groups and the opening of the E115^{RnfE}-R96^{RnfE}-E88^{RnfA} ion-pair network (Supplementary Fig. 20b,c,g and see Supplementary Fig. 9c), which could assist in pulling the Na⁺ from the inward to outward vestibules within the buried binding site (Fig. 4d, e, Supplementary Fig. 20d,e,h). The 2D free energy profiles suggest that the free energy barrier is rather small ($\Delta G^\ddagger = +4 \text{ kcal mol}^{-1}$) and the process is nearly isoenergetic when the AE1 is reduced (Fig. 4d). However, in stark contrast, when the AE1 cluster is oxidised, the process becomes endergonic ($\Delta G^\ddagger = +8 \text{ kcal mol}^{-1}$), and the reaction has a significantly higher free energy barrier ($\Delta G^\ddagger = +10 \text{ kcal mol}^{-1}$), thus supporting that reduction of the AE1 is coupled to the sodium translocation (Fig. 4d, Supplementary Fig. 19). The redox state dependence is also supported by solvation free energy calculations of the Na⁺ ion along the conduction pathway, suggesting that reduction of the AE1 stabilises the Na⁺ binding by 1-4 kcal mol⁻¹ along the pathway (Supplementary Fig. 15f). These results support our general mechanistic proposals of the redox-driven conformational switching coupled to ion transport, but we note that future work is required to fully characterise the complete free energy landscape and how this process drives the ion transport across the membrane."

Fig. 4 [...] (d) 2D free energy profiles of the inward-outward transition coupled to the Na^+ translocation with AE1 reduced (left) or oxidised (right). RC_1 –distance between TM4 of RnfA and RnfE; RC_2 –distance of Na^+ and the centre of mass of RnfA/E, projected onto the Z-axis (see Supplementary Figs. 18-20 for details). (e) Structures along the free energy profile with AE1 reduced: RnfA/E (top), closeup of the Na^+ , AE1, and residues interacting with Na^+ (bottom). Left: inward state; middle: transition state (TS); right: outward state.

Additions in the SI:

Supplementary Fig. 18 | *Steered molecular dynamics exploration of the inward/outward transition.* (a) Overview of the reaction coordinates (R_1 and R_2) used for exploring the inward to outward transition of the RnfA/E subunits by steered molecular dynamics (SMD) simulations. R_1 was defined as the Ca mean distance between helices g_1 (residues $Q85^{\text{RnfA}}$ to $T110^{\text{RnfA}}$) and g_2 (residues $C108^{\text{RnfE}}$ to $A118^{\text{RnfE}}$), while R_2 was defined as the Ca mean distance between helices g_3 (residues 80^{RnfE} to 105^{RnfE}) and g_4 (residues 116^{RnfE} to 126^{RnfE}). See Supplementary Methods for details. (b) The reaction coordinate R_1 (blue) and R_2 (orange) during the SMD simulations. (c) Distance of the Na^+ ion to the centre of the RnfA/E subunits, projected on the Z-axis shows how the ion moves from the intracellular side to the extracellular side during the SMD simulation. (d) Snapshots from the SMD simulation, showing the Na^+ ion and coordinating residues at different timepoints.

Supplementary Fig. 19 | Free energy calculations of the inward-outward transition. (a, b) 2D reaction coordinates (RC) used for the string simulations. (a) RC_1 is defined as the linear combination of two distances $R_2 - R_1$. R_1 is the mean distance between Ca atoms of helices g_1 (residues $85^{RnfA} - 110^{RnfA}$) and g_2 (residues $108^{RnfE} - 118^{RnfE}$), while R_2 is the mean distance Ca atoms of helices g_3 (residues $80^{RnfE} - 105^{RnfE}$) and g_4 (residues $116^{RnfE} - 126^{RnfE}$). (b) RC_2 is defined as the distance between the Na^+ ion and centre of mass (CoM) of RnfA/E Ca atoms projected onto the Z-axis. (c) Projection of the SMD trajectory onto the 2D reaction coordinates RC_1 and RC_2 (see Supplementary Fig. 18 and Supplementary Methods). Initial structures for the string simulation windows were chosen from the SMD simulation by equidistant spacing in the RC_1 dimension. (d, g) Optimisation of the string pathway with a (d) reduced or (g) oxidised AE1-cluster. The initial pathway (iteration 0) of the oxidised state was started from the fifth string iteration of the reduced state. (e, h) 2D Free energy landscape of the (e) reduced and (g) oxidised state simulations. (f, k) Phase space sampling of the converged string in (f) the reduced and (k) oxidised states. (l, m) Sum of squares (blue) and maximum deviation (orange) between each string pathway iteration of (l) reduced and (m) oxidised states.

Supplementary Fig. 20 | Characterisation of structures and sampling along the string simulations (a) Snapshots from the converged string simulation, showing the transport of a Na^+ ion from the intracellular (top left) to the extracellular side (bottom right) during the inward/outward transition. Inset: closeup of the Na^+ coordination by the backbone of residues T111^{RnfA}, V106^{RnfE}, and L103^{RnfE}. The reaction coordinates (RC_1 and RC_2) from each snapshot are given above the inset. **(b, c)** Distance of the E115^{RnfE}-R96^{RnfE} ion pair during the string simulation for **(b)** the reduced and **(c)** oxidised states shows that the ion pair is open in the inward state, while it closes in the outward state. **(d, e)** Angle of the T111^{RnfA} backbone carbonyl projected onto the Z-axis during the string simulation for the **(d)** reduced and **(e)** oxidised states. The carbonyl moiety flips as the RnfA/E subunit changes its conformation and transports the Na^+ ion. **(g, h)** E115^{RnfE}-R96^{RnfE} ion pair distance **(g)** and T111^{RnfA} backbone carbonyl angle **(h)** from unbiased MD simulations (see Supplementary Table 6).

Supplementary Fig. 15 | [...] (f) Sodium binding free energy (from PBSA/MM) along the sodium pathway across the RnfA/E subunits, sampled during the free energy calculations of the inward-outward transition (see Supplementary Figs. 18-20), with different redox states of the AE1 cluster (red: reduced; blue: oxidised). The solid lines/points represent the block average along

the reaction coordinate, with a bin width of 1 Å (RC_2 , defined as the distance between sodium and the centre of mass of RnfA/E, projected onto the Z-axis, with $RC_2 > 0$ being the intracellular side, and $RC_2 < 0$ the extracellular side, see also Supplementary Fig. 18).

The detailed methods for the SMD, string simulations are described in the Supplementary Methods section.

Question: In the model, Fig. 4 energy profile, the ET driving Na^+ translocation is uphill in energy. What is the driving force for Na^+ pumping (against 180 mV on the membrane) then?

Answer: In the model, the process: $AE1^{red} + Na^+_{in} \rightarrow AE1^{red}:Na^+_{in} \rightarrow Na^+_{out} + FMN_G^{red}$ is -0.05 eV at 0 mV and +0.08 eV in 180 mV. These values are based on the relative location of the reactions within the membrane (see Supplementary Method, Free energy profiles for Na^+ transport). It is possible to have local uphill steps as long as the overall process is favourable.

Question: The idea that the redox cycling of FeS in EA can be involved in Na^+ translocation is clear and appealing; however, the current evidence seems to be only partial, this should be more clearly stated in the text.

Answer: We thank the reviewer for the suggestion. Indeed, our results suggest that reduction of AE1 stabilizes sodium binding to the RnfA/E subunits. We agree that further evidence is also needed to elucidate the mechanism.

“However, the Rnf complex employs a complete inward-outward transition (Supplementary Fig. 11, Supplementary Movie 3-5) to establish the barrier modulations, as suggested by analysis of the cryo-EM data and the global motions observed in our MD simulations, thus further supporting a rocker-switch type alternative access mechanism. Our results suggest that Na^+ translocation is driven by the redox state of AE1, whilst further evidence to support our mechanistic model is also needed in future studies.”

Minor:

Question: I suggest to comment on what “thermodynamic limit of life” is in Intro.

Answer: We thank the reviewer for the suggestion and included a short clarification on the introduction:

“In many anaerobic bacteria and archaea that grow at the thermodynamic limit of life, where oxidation of low energy substrates result in small driving forces³, the generation of ion gradients relies on the membrane-bound Rnf complex⁴”

Question: What are the charges of FeS clusters in MD simulations? The data should be included in SI. How do we know their redox potentials? Same for FAD's.

Answer: We apologise for the missing data. We have now clarified the redox/charge state of the cofactors, and included the RESP charges for the iron-sulphur clusters and the flavin cofactors in a new Supplementary Table 10.

Supplementary Table 10 / RESP charges of iron-sulphur clusters, FMN and riboflavin cofactors. Four different FMN redox states are shown; FMN (oxidised, total charge = -2), FMN^{•-} (reduced, 1e⁻, total charge = -3), FMNH[•] (reduced, 1H⁺/1e⁻, N5 protonated, total charge = -2), and FMNH⁻ (reduced, 1H⁺/2e⁻, N5 protonated, total charge = -3). Five different riboflavin (RBF) redox states are shown; RBF⁻ (oxidised, total charge = 0), RBF^{•-} (reduced, 1e⁻, total charge = -1), RBFH[•] (reduced, 1H⁺/1e⁻, N5 protonated, total charge = 0), RBFH⁻ (reduced, 1H⁺/2e⁻), RBFH₂ (reduced, 2H⁺/2e⁻, N2 and N5 protonated, total charge = 0). RBF⁻ bonded terms are from CHARMM36 cgenff parameters, while the charges were calculated using restrained electrostatic potential (RESP) at B3LYP-D3 / def2-TZVP / ε = 4 level.

4Fe4S			2Fe2S		
State	Oxidised 1Fe ^{III} /3Fe ^{III}	Reduced 2Fe ^{II} /2Fe ^{III}	State	Oxidised Fe ^{III} /Fe ^{III}	Reduced Fe ^{II} /Fe ^{III}
Atom Name	Charge		Atom Name	Charge	
1FE1	0.797136	0.900785	1FE1	0.68059	0.74897
1FE2	0.761617	0.888277	1FE2	0.679	0.72132
1FE3	0.761617	0.900785	1S1	-0.66354	-0.84011

[...]

Question: What are the conclusions from ET calculations detailed in the extended Table 8, are there correlations with experiment? Some ET are likely gated or involve PT, this would make rather drastic changes I think to the formal rates shown.

Answer: The values represented in the table are based on experimental redox potential of the same cofactors measured in similar systems, whereas we have taken into account the *edge-to-edge* distances observed during our MD simulations. Indeed, the conformational state, influenced by the redox state, gates the electron transfer by modulating distances involving e.g. B8-AE1 and AE1-FMN^{RnfG}. There is currently no available experimental rate constant for this system, so the model cannot be directly compared to experiments. From these cofactors, only the reduction of NAD⁺ to NADH involves a PCET reaction, that has been taken into account on the rate constant based on calculations from a previous study.

Comment: Despite many questions, it's a great work, congratulations.

Answer: We wish to thank the reviewer for the encouraging comments.

Reviewer #4 (Remarks to the Author):

In their paper titled “Molecular principles of redox-coupled sodium pumping of the ancient Rnf machinery” Kumar and coworkers present high resolution cryo-EM structures of the Rnf complex from *A. woodii* in a NADH- and a Fd-reduced state. They present all-atom molecular dynamics simulations and biochemical data to explain the elusive mechanism underpinning the redox-driven Na⁺ transport. Characterization of Rnf has far-reaching implications in biochemistry, energy metabolism, and biotechnology, therefore the manuscript is both timely and relevant with potential for high impact.

Overall, the manuscript is well-organized, the conclusions are supported by the data, and the uncertain steps of the mechanism are mentioned with the appropriate care. However, the paper would benefit from discussing the existing theories in more detail, specifically in the introduction.

Answer: We thank the reviewer for the positive comments and appreciation of our work.

We highlight in the Introduction that there is currently no consensus about the mechanism, in particular, the location of the Na⁺ pumping pathways. However, since we do not discuss structural details in the Introduction section, we wish to present the mechanistic models later in the Discussion section:

“However, despite extensive structural studies of both Rnf¹² and Nqr^{1,13}, the mechanism by which the electron transfer is coupled to sodium pumping remains elusive and highly debated, with several conflicting suggestions on the location of the ion translocation pathways.”

Here, as per my expertise, I focus on the assessment of the reported simulations. These simulations provide basis for the majority of the findings presented in the manuscript, and, when possible, are cross-checked against the cryo-EM data, and complemented by mutational studies and enzymatic activity measurements.

Question: The simulations were initiated either from the NADH- or Fd-reduced Rnf models with unspecified numbers of water molecules and POPC lipids. This is also difficult to judge from Extended Data Fig. 7/b. I recommend adding these numbers to the relevant Extended Data Methods section along with a brief justification of the higher-than-physiological salt concentration.

Answer: We thank the reviewer for the suggestions. We have included the requested MD simulation data in the Supplementary Methods section.

“The full systems comprised ca. 430,000 atoms, with the NADH-reduced model including 100,572 water molecules, 506/482 Na⁺/Cl⁻ ions, and 448 POPC molecules, and the Fd-reduced model containing 115,908 water molecules, 566/553 Na⁺/Cl⁻ ions, and 448 POPC molecules. The MD simulation box dimensions were 170x164x180 Å and 166x158x208 Å for the NADH- and Fd-reduced models, respectively.”

We use a higher salt concentration in the simulations to increase the likelihood of Na⁺ binding to the protein within the accessible simulation timescales. This has now been clarified in the Supplementary Methods section:

“Na⁺ and Cl⁻ ions were added to neutralise the system with a salt concentration of 250 mM, higher than the physiological value (150 mM) to increase the likelihood of Na⁺ binding to the protein.”

Question: The 500 ns-long (except S15 to S20) simulations follow the standard protocol and were performed in duplicates. The trajectories were classified into outward-open and inward-open states, from which the authors showed that the NADH-reduced simulations always remain in the outward-open state while some of the Fd-reduced runs show transitions between the two states. However, there is but one combination where the simulations were identical in terms of the reduced cofactors and differed only in the initial structure, which is S5,S6 and S9,S10. To support this finding, I recommend the authors repeat the simulations with S13,S14 starting from the NADH-reduced structure. Performing these additional simulations is by no means a large computational burden and will strengthen the conclusions of the manuscript.

Answer: We thank the reviewer for the suggestion. We have now completed the new simulations (S7-S12, see Supplementary Table 6), starting from the NADH-reduced state. Our extended data show similar trends as in the previous simulations, further supporting the general conclusions. We have also extended the simulations S13-S20, S27, and S28 to 1 μ s, as well as extending the simulations S21-S26 to 200ns, to further strengthen the statistical significance of the results.

Finally, I detail below a list of smaller inaccuracies and typographical errors I noticed in the manuscript:

Question:

- L127: the termini of RnfC are discussed in the text but not shown in Fig.1/c.

Answer: This has been now included in the revised figure.

Question:

- L203: where does the B8 domain insert into? The figures suggest the cytoplasmic funnel, but this is not written.

Answer: We apologise for the missing information. We have revised the sentence:

“These conformational changes are enabled by the insertion of the B8-domain (residues 30-100 of RnfB, see Supplementary Fig. 7e, f) *between the RnfA/E subunits* together with conformational changes in TM1/4 of RnfA.”

Question:

- L313: “(the Na⁺ interaction with the site)” should read “(due to the Na⁺ interaction with the site)”

Answer: The sentence has now been clarified:

“our calculations suggest that the redox potential of AE1 downshifts by ca. 130 mV (due to the Na⁺ interaction with the site) that favours the electron transferred to FMN^G”

Question:

- L404: the “proposed mechanism” and the “fundamental principle” are not stated explicitly

Answer: We have revised the sentence to clarify the message:

“The proposed *redox-driven alternate-access* mechanism introduces a novel principle for ion pumping employed by the Rnf and related enzymes that appeared early on during evolution to facilitate survival under energy-limited conditions at the thermodynamic limit of life. Nevertheless, *similar molecular principles could be* also employed in contemporary aerobic organisms, where the related Nqr is utilised as a respiratory machine in many important pathogens. The proposed redox-driven sodium pumping mechanism could thus provide a basis for drug design to combat emerging pathogens.”

Question:

- L779: the eye-symbol in Fig.3/g is not used and not mentioned.

Answer: The clarification has been added to the caption.

“Structural snapshots from MD simulations (S15, S16), showing sodium binding in the intracellular (e) and extracellular (f) buried binding sites within RnfA/E (green/red) near the AE1 cluster. The binding is stabilized by backbones of V106^{RnfA}, L103^{RnfE}, and T111^{RnfA}, as well as by N112^{RnfA} side chain in the intracellular side. The perspective for the MD snapshots is indicated in panel g.”

Question:

- L782: contrary to the caption, Fig.3/g is a density profile and not a radial distribution function

Answer: We apologise for the confusion. We have revised the figure caption:

“Probability density of sodium ion from the RnfA/E (left) and RnfD (right) centres during the MD simulations.”

Question:

- L880: incorrect reference to Extended Data Fig.1/a, should be 7/a

Answer: The figure reference has now been corrected (now Supplementary Figure 8a).

“The FMN cofactors, located in RnfG and RnfD subunits, were modelled by covalently linking them to T185^{RnfG} and T156^{RnfD} via a phosphodiester bond (Supplementary Fig. 8a).”

Question:

- L1076 & L1107: unresolved abbreviation of ISC

Answer: The abbreviation has been clarified.

“Supplementary Fig. 8 | MD simulations of the Rnf complex. (a) Structural overview of the Rnf system used for MD simulations. Subunits RnfA (teal), RnfB (purple), RnfC (blue), RnfD (pink), RnfE (red), RnfG (green) are shown in cartoon representation. The iron-sulphur clusters are represented as spheres (sulphur:yellow, iron:orange), while the cofactors FMN and RBF are shown as sticks.”

“Supplementary Fig. 10 | Summary of cofactor distances from MD simulations. (a) Structural overview of the Rnf showing the location of the cofactors after MD equilibration (S15). Inset: A snapshot from MD simulation S15 showing the flexible domain of RnfB, containing the B8 cluster, binding close to the membrane subunits of RnfA and RnfE. Two positively charged residues (Lys32^{RnfA}, Arg154^{RnfA}) are involved in electrostatic interaction with the iron-sulphur cluster.”

Question:

- L1084: the mentioned RMSF data seems to be missing

Answer: We apologise for the mistake and the caption has been corrected. The RMSF analysis is shown in Supplementary Fig. 5 g.

“(g) Average root-mean-square fluctuations (RMSF) calculated from the MD simulations and mapped on the cryo-EM structure of the Rnf complex treated with reduced Fd.”

Question:

• L1097: the sentence “(f) Angle of backbone... projected onto Z-axis (in Å)...” does not make sense, and the associated panel shows the angle and no distances.

Answer: We apologise for the mistake and the unit has been changed to degrees (°) in the caption and the sentence clarified.

“(f) Angle *between the* backbone carbonyls from V106^{RnfE} (top) and T111^{RnfA} (bottom) *and the* Z-axis (in °) from MD simulations with a reduced AE1 cluster (S13-S16, see Supplementary Table 6).”

Question:

• Extended Data Fig.10 : RnfB should be RnfE.

Answer: The label has now been fixed.